# Biochemical mechanisms determine the functional compatibility of heterologous genes

Andreas Porse[1], Thea S. Schou[1], Christian Munck[1], Mostafa M.H. Ellabaan[1] & Morten O.A. Sommer [1]

Elucidating the factors governing the functional compatibility of horizontally transferred genes is important to understand bacterial evolution, including the emergence and spread of antibiotic resistance, and to successfully engineer biological systems. In silico efforts and work using single-gene libraries have suggested that sequence composition is a strong barrier for the successful integration of heterologous genes. Here we sample 200 diverse genes, representing >80% of sequenced antibiotic resistance genes, to interrogate the factors governing genetic compatibility in new hosts. In contrast to previous work, we find that GC content, codon usage, and mRNA-folding energy are of minor importance for the compatibility of mechanistically diverse gene products at moderate expression. Instead, we identify the phylogenetic origin, and the dependence of a resistance mechanism on host physiology, as major factors governing the functionality and fitness of antibiotic resistance genes. These findings emphasize the importance of biochemical mechanism for heterologous gene compatibility, and suggest physiological constraints as a pivotal feature orienting the evolution of antibiotic resistance.

[1] Novo Nordisk Foundation Center for Biosustainability, Technical University of Denmark, Kgs. Lyngby, DK-2800, Denmark. Correspondence and requests for materials should be addressed to M.O.A.S. (email: msom@bio.dtu.dk)

A distinct feature of prokaryotes is their ability to share genetic material via horizontal gene transfer[1]. Such open-source evolution provides rapid access to the genetic innovations that continuously shape bacterial genomes[2]. The transfer of genes between bacteria happens primarily via transferrable genetic elements such as plasmids, phages, or direct DNA uptake (transformation)[3]. Although such transfer events are believed to occur frequently in nature, the fundamental forces governing the establishment and maintenance of successfully transferred genes are poorly understood[2,4]. Because foreign genes may lose functionality or impose a high biological cost in new hosts, this gap in our understanding also limits current synthetic biology efforts focused on engineering novel functions into biological systems[5,6].

From computational studies of sequenced genomes, the tendency of a gene to be transferred has been inferred to depend mainly on ecological and phylogenetic factors[1,4]. As physical proximity of donor and recipient bacteria is generally required for transfer, ecology has been suggested as a strong dissemination barrier[4]. However, the broad host range of some transfer mechanisms, and the ubiquitous presence of antibiotic resistance genes across environments, suggest that ecological barriers are largely governed by functional constraints[7,8]. Indeed, numerous studies reveal a functional bias of transferred gene categories, with genes encoding largely independent products, e.g., those involved in secondary metabolism or virulence, being transferred more often than genes encoding highly interactive proteins involved in transcription and translation[9–12]. Furthermore, metagenomic analyses of DNA from different environments suggest that genes involved in antibiotic resistance are more confined by phylogeny, indicating that genomic factors are important for the acquisition or maintenance of foreign genetic material[4,13,14]. In support of this idea, in silico studies have shown a bias in the nucleotide composition of transferred genes in relation to the recipient genome, and suggest that sequence parameters influence successful gene integration[15–17]. Specifically, a role of the codon usage and GC content on the functional integration of acquired genes has been inferred due to the potential role in gene expression and fitness[18,19]. Upon successful transfer, host compatibility and selection are crucial for the long-term persistence of newly acquired genetic material, as costly genes will eventually be lost through purifying selection[20–25].

Experimental studies characterizing the phenotypic effects of synonymous sequence variation have largely focused on a limited set of phenotypes and most current data are obtained from fluorescent protein expression[26–28]. However, recent work by Kacar and Garmendia et al. showed an increased fitness cost of replacing the elongation factor Tu (a highly conserved protein involved in translation) with its distant homologs in *Escherichia coli*, and similar results have been obtained by Lind et al. when replacing ribosomal subunits in *Salmonella typhimurium*[29,30]. While the fitness cost of expressing these core translational genes could not be attributed to differences in sequence composition, Amorós-Moya et al. showed that sub-optimal codon usage of a highly expressed chloramphenicol acetyl transferase resistance gene resulted in lower resistance levels and a decrease in overall host fitness[31]. Although existing studies provide interesting clues on the relation among sequence composition, gene expression, and fitness, the relevance of these findings regarding the functionality of diverse naturally occurring accessory genes involved in antibiotic resistance is not clear.

Antibiotic resistance genes are ubiquitously present on mobile genetic elements that allow their extensive dissemination among bacteria[3]. Given the stagnation in antibiotic discovery, the increasing prevalence of multidrug resistant bacteria constitutes an urgent threat to public health that motivates a deeper understanding of the forces governing the dissemination and long-term maintenance of antibiotic resistance genes[32–34].

Antibiotic resistance is conferred through five major mechanisms: (i) enzymatic drug inactivation, (ii) active drug efflux, (iii) modification of drug target, (iv) replacement of the drug target with a resistant variant, and (v) regulatory shifts towards a more resistant phenotype[35]. The diversity of mechanisms, through which antibiotic resistance is achieved, makes antibiotic resistance genes a valuable model system for investigating the factors that may affect the functional compatibility of transferable genes in general.

In this study, we employ a synthetic bottom-up approach by sampling a broad sequence space of 200 diverse open reading frames annotated as antibiotic resistance genes. Via experimental profiling of these genes, we discover that resistance mechanisms and the phylogenetic relatedness of donor and recipient species act together as important determinants of gene functionality and fitness cost. Consequently, we suggest that these effects dominate the potential transfer barriers imposed by sub-optimal sequence composition of heterologous genes.

## Results

**200 genes representatively sampled from major databases.** By clustering all genes of major publicly available antibiotic resistance gene databases, and selecting the most abundant genotypes, we obtained 200 genes for DNA synthesis (Methods, Supplementary Fig. 1a and Supplementary Data 1). The selected clusters represented the most abundant genes in resistance gene databases (>80%, Supplementary Fig. 1a) and accounted for 98% of the total data set homologs in general sequence databases (NCBI NT and Genomes) (Supplementary Fig. 1b).

The 200 selected genes vary widely in resistance mechanisms, targeted drug classes, and phylogenetic dissemination (Fig. 1, Supplementary Data 1). In total, 64% of the selected genes were associated exclusively with Gram-negative organisms, and of the remaining genes, 13% were found in Gram-positive organisms, and 10.5% were found in both Gram-negative and Gram-positive organisms (Fig. 1c). However, 12.5% of the genes had not been associated with a particular host organism (mostly genes found via metagenomic functional selections). The selected genes were annotated to confer resistance to 11 distinct drug classes via 23 distinct biochemical functions that can be divided into five major mechanistic categories based on their Resfam annotations (Fig. 1a, b)[36]. In addition to genes annotated to confer resistance towards known drug classes, genes annotated to confer antibiotic resistance but without defined antibiotic resistance profiles were included as a consequence of their high abundance in the public antibiotic resistance databases. All of these genes were annotated as having regulatory or efflux functions but were not associated with specific drug classes in the literature (Supplementary Data 1).

**Functional characterization of putative resistance genes.** The 200 selected genes were cloned in a low-expression setup in *E. coli* MG1655 (Supplementary Fig. 1). All genes were subjected to 20 phenotypic tests and growth rate measurements. Specifically, the resistance towards 20 antibiotics comprising 12 chemical classes was assessed. A gene was considered a functional resistance gene if it conferred a resistance phenotype of at least a twofold increase in the minimal inhibitory concentration (MIC) compared to that of *E. coli* MG1655 carrying the empty vector (Supplementary Table 1). Clones displaying a resistance phenotype were subjected to growth rate assessments under non-selective conditions and further antibiotic susceptibility testing in a range from 2- to 30-fold the WT MIC (Fig. 2).

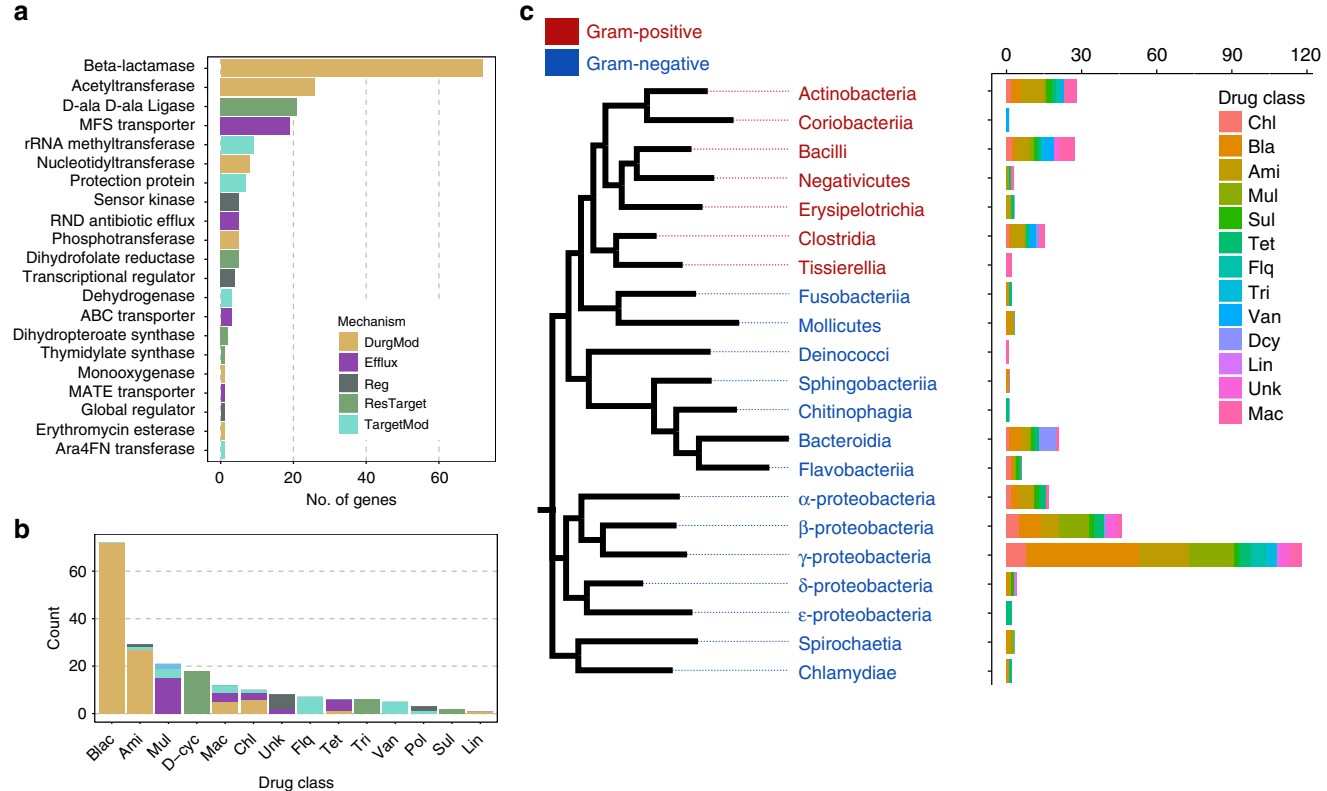

**Fig. 1** Database mechanistic and phylogenetic diversity of 200 synthesized genes. **a** Biochemical functions of the included antibiotic resistance genes obtained via Resfams. These genes are grouped into five major mechanistic categories (relative abundance shown in parenthesis): DrugMod: drug inactivating enzymes (56.5%), Efflux: efflux pumps (14%), Reg: regulators (5%), ResTarget: redundant/resistant target (14.5%), TargetMod: target modifying/binding proteins (10%). **b** Gene counts stratified by resistance class and fractionated by resistance mechanisms. **c** Phylogenetic and drug class distributions of the included genes. Genes that could be identified (97% identity) in one or more genomes deposited in RefSeq were quantified for each bacterial phylogenetic class. The coloring of each bar depicts the distribution of the annotated resistance. Drug class aberrations: Chl: Chloramphenicol, Bla: β-lactams, Ami: Aminoglycosides, Mul: Multiple drug classes, Sul: Sulfamethoxazole, Tet: Tetracyclines, Flq: Fluoroquinolones, Unk: Unknown, Tri: Trimethoprim, Van: Vancomycin, Dcy: D-cycloserine, Lin: Lincosamides, Mac: Macrolides, Pol: Polymyxins

Whereas only 32% of the 200 tested genes could be identified in *E. coli* genomes deposited in NCBI's RefSeq database, 63% of the genes displayed at least one resistance phenotype in *E. coli*. The resistance phenotypes were distributed unequally among the drug classes, with genes annotated to confer resistance towards tetracyclines, sulfonamides, β-lactams, and fluoroquinolones having a high proportion (>80%) of functional variants. By contrast, D-ala ligases, which confer resistance to D-cycloserine in *E. coli*, as well as genes annotated to confer resistance towards polymyxins or multiple drug classes, showed the lowest proportion of functional genes in our experimental setup (Fig. 2, Supplementary Data 1). Genes conferring resistance towards β-lactams, aminoglycosides, chloramphenicol, and trimethoprim showed the highest average levels of resistance, whereas those conferring resistance towards fluoroquinolones and D-cycloserine displayed the lowest average increase in resistance compared to the susceptible WT (Fig. 2). This difference in resistance level between drug classes was statistically significant (ANOVA, $P <$ 0.001). We further investigated if the level of resistance could be attributed to differences in the codon adaptation index (CAI), GC content or mRNA-folding energy; however we could not detect any significant correlations between these sequence parameters and the resistance level for the total data set nor within resistance classes (Supplementary Fig. 3).

**Sequence composition is not a major functional barrier.** As 68% of the tested genes from our data set have not yet been identified in sequenced *E. coli* genomes, our data set is well suited

for studying factors relevant to the functionality of resistance genes evolved in a foreign genetic context. The selected genes varied widely in their base composition (Fig. 3), which has previously been hypothesized to affect functional expression and successful gene transfer[16,18,19].

The codon usage of an incoming gene might influence its protein expression and it is generally believed that the CAI is important for heterologous gene integration[18,26,37,38]. Surprisingly, we found that the average *E. coli* CAI of functional resistance genes was slightly lower compared to that of the non-functional genes (Mann–Whitney *U*-test, $P = 0.014$; Fig. 3a). Yet, we found no significant difference in the average GC content between the functional and non-functional genes (Mann–Whitney *U*-test, $P = 0.12$) (Fig. 3b). To investigate whether interactions between these and other parameters would influence the outcome of our analysis, we built a multivariate logistic regression model (Supplementary Table 2). The inclusion of GC content did not significantly change the predictive power of the model compared to CAI alone ($P = 0.70$), suggesting that the effect of the CAI was not governed by the GC content (Fig. 3d). The folding energy of the N-terminal of a transcript may also influence gene expression[26,28,39], however, N-terminal mRNA-folding energy did not predict functionality, nor resistance level of the functional genes, in our logistic regression model (Fig. 3c, Supplementary Fig. 3 and Supplementary Table 2).

Although gene functional compatibility exhibited little dependence on sequence composition, it may affect the fitness cost in

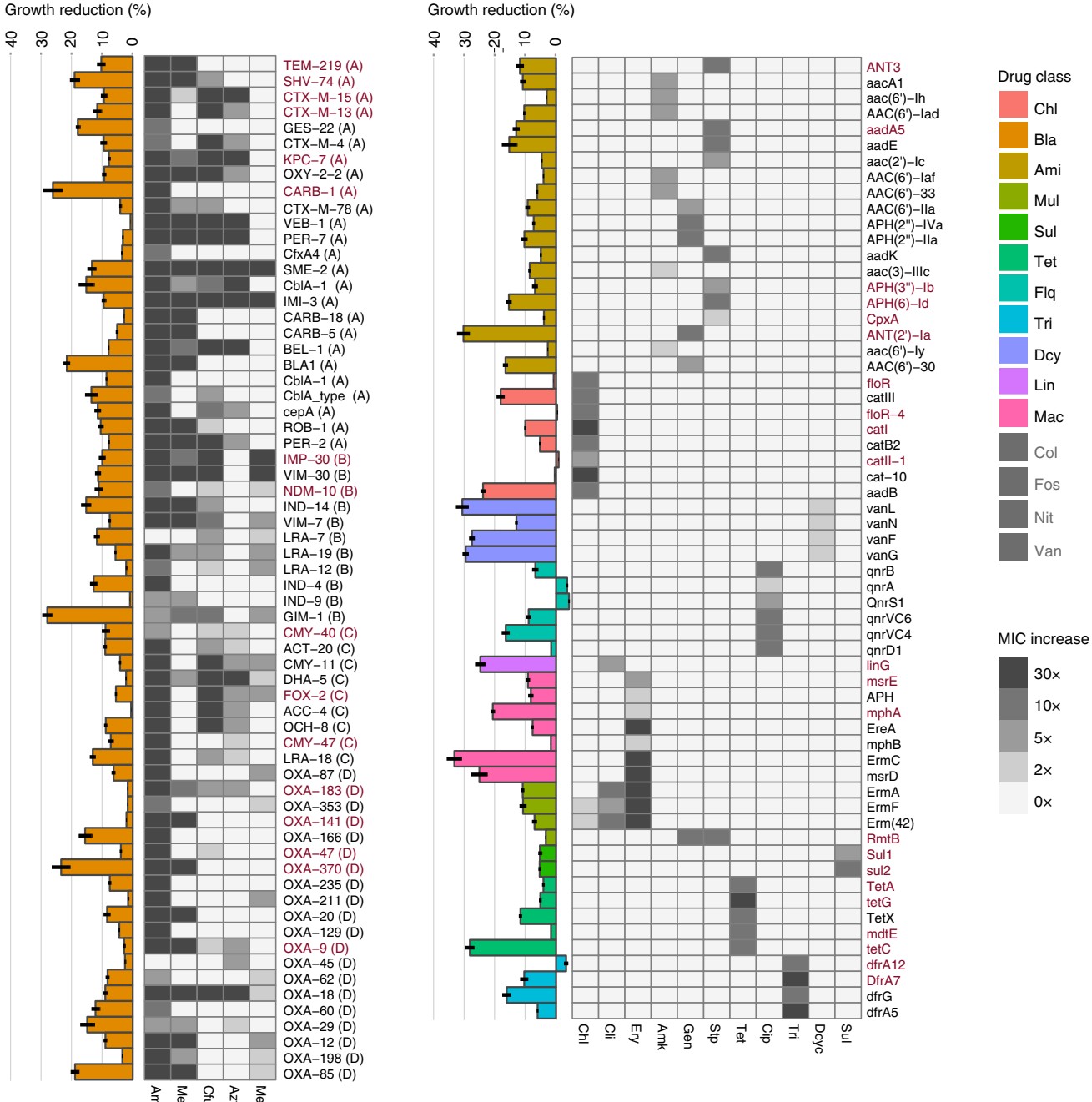

**Fig. 2** The resistance level and relative growth rate of each functional resistance gene in *E. coli*. The heatmaps show the resistance profile (fold change in MIC compared to *E. coli* MG1655 carrying the empty expression backbone) for all functional resistance genes (*n* = 126) and are grouped by drug class. The β-lactamases (left) are clustered by their molecular (Ambler) classes as shown in parentheses[60]. Bars represent the mean of at least three repeated growth measurements and are normalized to *E. coli* MG1655 carrying the empty expression vector. Error bars show the standard error of the mean (SEM). The genes highlighted in red are present in sequenced *E. coli* genomes deposited in NCBI's RefSeq genome database. The drug classes are as follows: Chl: Chloramphenicol, Bla: β-lactams (Amx: Amoxicillin, Mec: Mecillinam, Cfu: Cefotaxime, Azt: Aztreonam, Mep: Meropenem), Ami: Aminoglycosides, Mul: Multiple drug classes, Sul: Sulfamethoxazole, Tet: Tetracyclines, Flq: Fluoroquinolones, Tri: Trimethoprim, Dcy: D-cycloserine, Lin: Lincosamides, Mac: Macrolides, Col: Colistin, Fos: Fosfomycin, Nit: Nitrofurantoin, Van: Vancomycin. Gray colored drugs were tested, but no genes conferred resistance towards these in *E. coli*

new hosts. As a proxy for fitness, we measured the growth rate of *E. coli* expressing each of the functional resistance genes (Fig. 2). Compared to *E. coli* carrying the empty expression vector, the impact of expressing a resistance gene on the growth rate ranged from a slight increase to more than a 30% decrease in growth rate (Fig. 2 and Supplementary Fig. 4).

The reduced growth rate resulting from expression of a resistance gene differed significantly between the drug classes to which the gene conferred resistance (one-way ANOVA, *P* > 0.001; Fig. 2), but the impact on growth did not differ between mechanistic categories (one-way ANOVA, *P* = 0.38; Supplementary Fig. 5). The gyrase-protecting *qnr* fluoroquinolone resistance

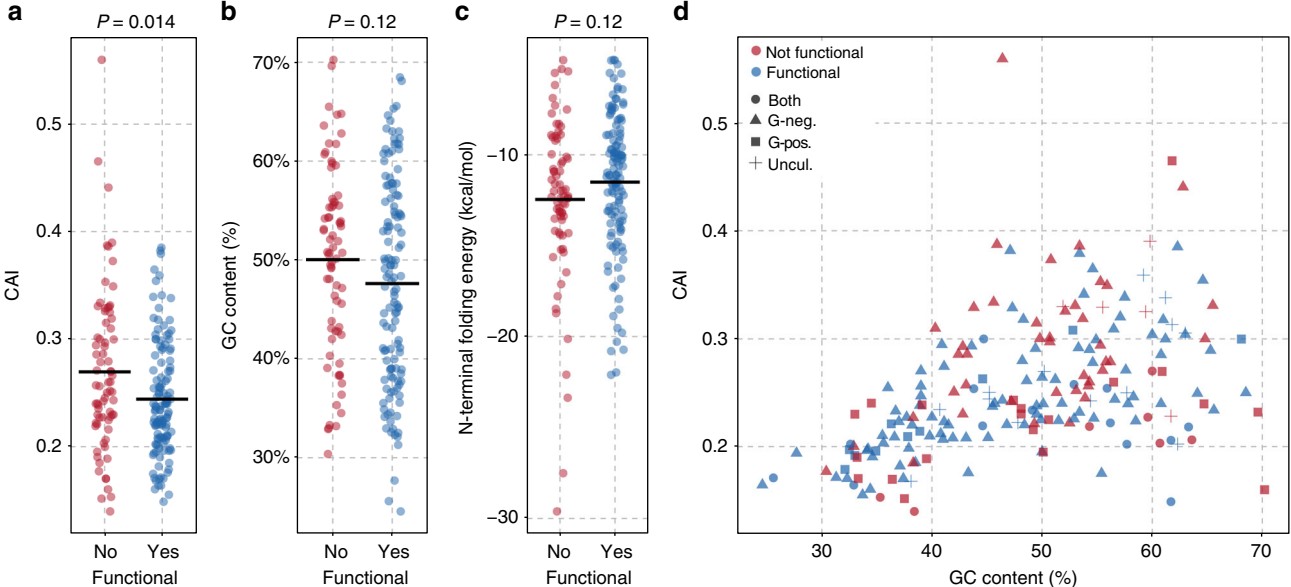

**Fig. 3** Divergent sequence composition is not a major barrier for functionality of a foreign antibiotic resistance gene in *E. coli*. **a** Comparison of the codon usage between functional and non-functional genes. **b** Comparison of the GC content between functional and non-functional genes. **c** Comparison of the mRNA N-terminal folding energy between functional and non-functional genes. **d** Functional and non-functional genes are dispersed throughout the sequence space. Point shape indicates whether a gene has been observed exclusively in Gram-negative organisms ($n$ = 132), Gram-positive organisms ($n$ = 30), in both Gram-positive and Gram-negative organisms ($n$ = 21), or has not yet been associated with a sequenced genome (uncultured, $n$ = 17)

genes showed exceptionally low costs compared to other resistance classes, especially the D-alanine–D-alanine ligases. These confer D-cycloserine resistance in *E. coli* through target replacement, and had a high negative impact on growth rates (Fig. 2).

Contrary to current thinking, some antibiotic resistance genes from the *qnr* and *dfr* families were slightly advantageous to *E. coli* in the absence of antibiotic selection (Fig. 2). We detected beneficial growth patterns of *qnrA* as well as *qnrS1*, which are found, for example, in *Shigella* and *Salmonella* species that are closely related to *E. coli*, but not for the *qnr* genes found in more distantly related species. Despite the observed trade-off between resistance level and growth rate for target-modifying genes ($r$ = 0.62, $P$ = 0.04), this was not the case for efflux mediators ($r$ = −0.59, $P$ = 0.12), resistant targets ($r$ = −0.61, $P$ = 0.06) and genes belonging to the drug-modifying mechanistic class ($r$ = 0.02, $P$ = 0.84; Supplementary Fig. 6).

It has previously been recognized that gene level parameters such as the CAI and GC content of codon-scrambled genes can influence host fitness when expressed in *E. coli*[26,27]. *E. coli* expressing the trimethoprim resistance gene *dfrG* of the dihydrofolate (*dfr*) family, originating from *Staphylococcus aureus*, displayed a lower growth rate compared to *E. coli* expressing *dfr* genes of Gram-negative origin (Fig. 2). Indeed, the *dfrG* gene had a low GC content (32%) and CAI (0.17), which we hypothesized could negatively influence the growth of *E. coli*. To test whether base composition could be optimized to decrease the growth reduction imposed by *dfrG*, we synthesized a codon-optimized variant with a higher GC content (55%) and CAI (0.66). However, this variant did not show a significant growth improvement compared to the wild-type *dfrG* gene when expressed in *E. coli* (Mann–Whitney *U*-test, $P$ = 0.1; Supplementary Fig. 7), suggesting that factors beyond the nucleotide sequence are responsible for the differential cost.

To assess the general trend in our data set, we investigated the correlations between growth rate effects and the CAI, GC content, N-terminal mRNA-folding energy, and gene length (Fig. 4). Although there was a tendency towards higher growth rates for

genes with an increasing CAI, in contrast to previous studies on sequence variants, including codon-scrambled *gfp*[26,27], we found no significant correlation between these individual sequence parameters and the growth rate of *E. coli* for our diverse set of antibiotic resistance genes.

A multiple linear regression model of our growth data against the CAI, GC content, N-terminal mRNA-folding energy, and gene lengths explained virtually no variation in the growth rate ($R^2$ = −0.01, $P$ = 0.6; Supplementary Table 3). This result indicated that other factors dominate the potentially minor growth effects imposed by the base and codon composition in our data set.

**Resistance mechanism is a major determinant of gene compatibility.** The functional compatibility of a gene in a new host may depend on its interaction with specific components of host physiology and metabolism[9]. Antibiotic resistance genes mediate their phenotype through a wide range of cellular interactions, with some mechanisms being dependent on specific host structures, e.g., ribosomal structure or the cell envelope for target modifying and efflux mechanism respectively, whereas drug modifying enzymes act directly on the antibiotic compound[35].

Although we observed a small significant difference in the proportion of functional genes for different targeted drug classes ($\chi^2$, $P$ = 0.031), we also noted that genes annotated to confer resistance to a specific drug class were frequently dominated by specific resistance mechanisms (Fig. 1b)[40]. Accordingly, the mechanistic category of a gene was far better at predicting the functionality of a gene transferred to *E. coli* ($\chi^2$, $P$ < 0.001) than the targeted drug class alone. This result suggested that certain resistance mechanisms are easier to integrate into a novel host physiology than others (Fig. 5a and Supplementary Table 2). The highest proportion of functional genes was found among the drug-modifying enzymes, including the β-lactamases and aminoglycoside transferases, with most genes conferring high levels of resistance (Fig. 5a), and this distribution was not significantly biased by the phylogenetic affiliation of these genes ($\chi^2$, $P$ = 0.08) or whether they had previously been identified in *E. coli* ($\chi^2$, $P$ = 0.77).

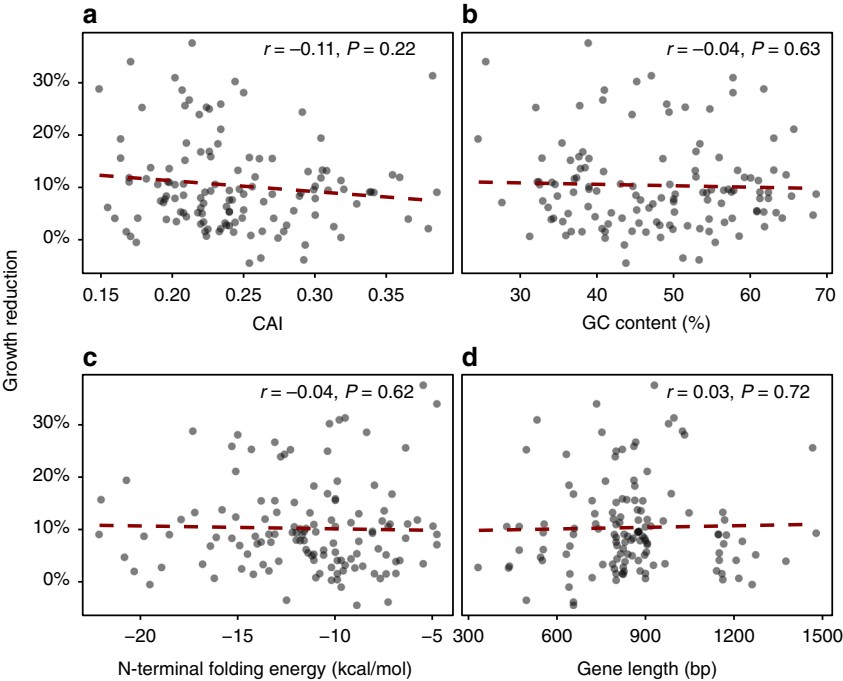

**Fig. 4** The cost of resistance gene acquisition shows little or no dependence on sequence composition. **a** The CAI showed a small ($r = −0.11$) but non-significant ($P = 0.22$) correlation with growth reduction. Similarly, the GC content ($r = −0.04$) (**b**), N-terminal mRNA-folding energy ($r = −0.04$) (**c**), and gene length ($r = 0.03$) (**d**) did not correlate significantly with the growth reduction experienced by *E. coli* upon receiving the resistance genes in our expression setup ($P > 0.05$)

By contrast, genes conferring resistance through efflux and regulatory mechanisms were least likely to function in *E. coli*, in which only 28.5% and 10% of genes displayed resistant phenotypes, respectively. These findings are consistent with the hypothesis that genes involved in limited cellular interactions are more likely to be functionally compatible in a new host[9].

**Phylogeny affects fitness of cell-interacting resistance genes**. If the activity of a gene product is detrimental for functional genetic integration due to suboptimal physiological interactions, we would expect the phylogenetic relatedness of the donor and recipient species to influence the functional compatibility and fitness cost of newly acquired genes. This hypothesis was supported by the fact that the deviation in GC content from the *E. coli* average (50.8% GC) was a stronger, although still not significant, predictor of the growth reduction resulting from gene expression ($r = 0.16$, $P = 0.07$; Supplementary Fig. 8) compared to the absolute GC content (Fig. 4b). As GC content varies among phylogenetic groups, this trend could be a proxy for the cellular environment in which a gene has evolved, rather than a direct effect of sequence composition (Supplementary Fig. 9). Indeed, a higher average growth rate of *E. coli* was observed for genes affiliated with Gram-negative organisms compared to those that were exclusively associated with in Gram-positive organisms or both (Supplementary Fig. 10. Mann–Whitney *U*-test, $P = 0.03$).

To further investigate the basis of the fitness costs of gene expression, we derived a distance measure based on the average 16S rRNA sequence identity between *E. coli* and the genomes in which the gene has been identified. Using this 16S rRNA-based evolutionary distance measure, we found a correlation between the cost of a gene and the relatedness of its genomic context to *E. coli* ($r = 0.29$, $P = 0.003$). Whereas the relative burden of expressing drug-modifying enzymes was independent of phylogenetic relatedness of the typical hosts of a gene to *E. coli* ($r = 0.04$, $P = 0.56$; Fig. 5c), we found the main drivers of this

correlation to be the genes that directly interact with cellular components ($r = 0.74$, $P < 0.001$; Fig. 5c). This correlation was not biased by differences in cell-interacting mechanistic categories (ANOVA, $P = 0.93$; Fig. 6), targeted drug classes (ANOVA, $P = 0.57$) or phylogenetic groups (ANOVA, $P = 0.23$; Supplementary Fig. 11). However, the effects were most pronounced within the target replacing and target modifying mechanistic classes, and for genes affiliated with Proteobacteria and Actinobacteria (Fig. 6 and Supplementary Fig. 11). These results suggests a trade-off between adaptation towards one host and being broadly functional across phylogeny, which is further supported by the higher cost of genes disseminated across Gram-positives and Gram-negatives (Supplementary Fig. 10).

While there was an unequal distribution of functional genes among Gram-classes ($\chi^2$, $P = 0.002$), the phylogenetic affiliation, measured as the average 16S rRNA-based distance between genomes harboring the gene, was not significantly correlated with gene functional compatibility when all genes in our data set were considered (Mann–Whitney *U*-test, $P = 0.84$). Although a bigger difference was observed when cell-interacting genes were considered in isolation, this difference was still not significant (Mann–Whitney *U*-test, $P = 0.23$; Fig. 5d). However, when excluding native *E. coli* genes (Supplementary Table 4), a highly significant dependence of functionality on phylogenetic distance for cell-interacting proteins (Mann–Whitney *U*-test, $P < 0.001$), but no change for drug-interacting resistance mediators (Mann–Whitney *U*-test, $P = 0.57$), was observed. For this subset of genes, the effect of phylogenetic differences was dominated by genes conferring resistance through target replacement (Mann–Whitney *U*-test, $P = 0.031$) and efflux mechanisms (Mann–Whitney *U*-test, $P = 0.015$). However, while the Gram-class affiliation was not a significant predictor of functionality for efflux, target replacing and regulatory genes ($\chi^2$, $P > 0.05$), genes originating from Gram-negatives were significantly overrepresented in functional target-modifying resistance mediators ($\chi^2$, $P = 0.005$). These observations further support the importance of

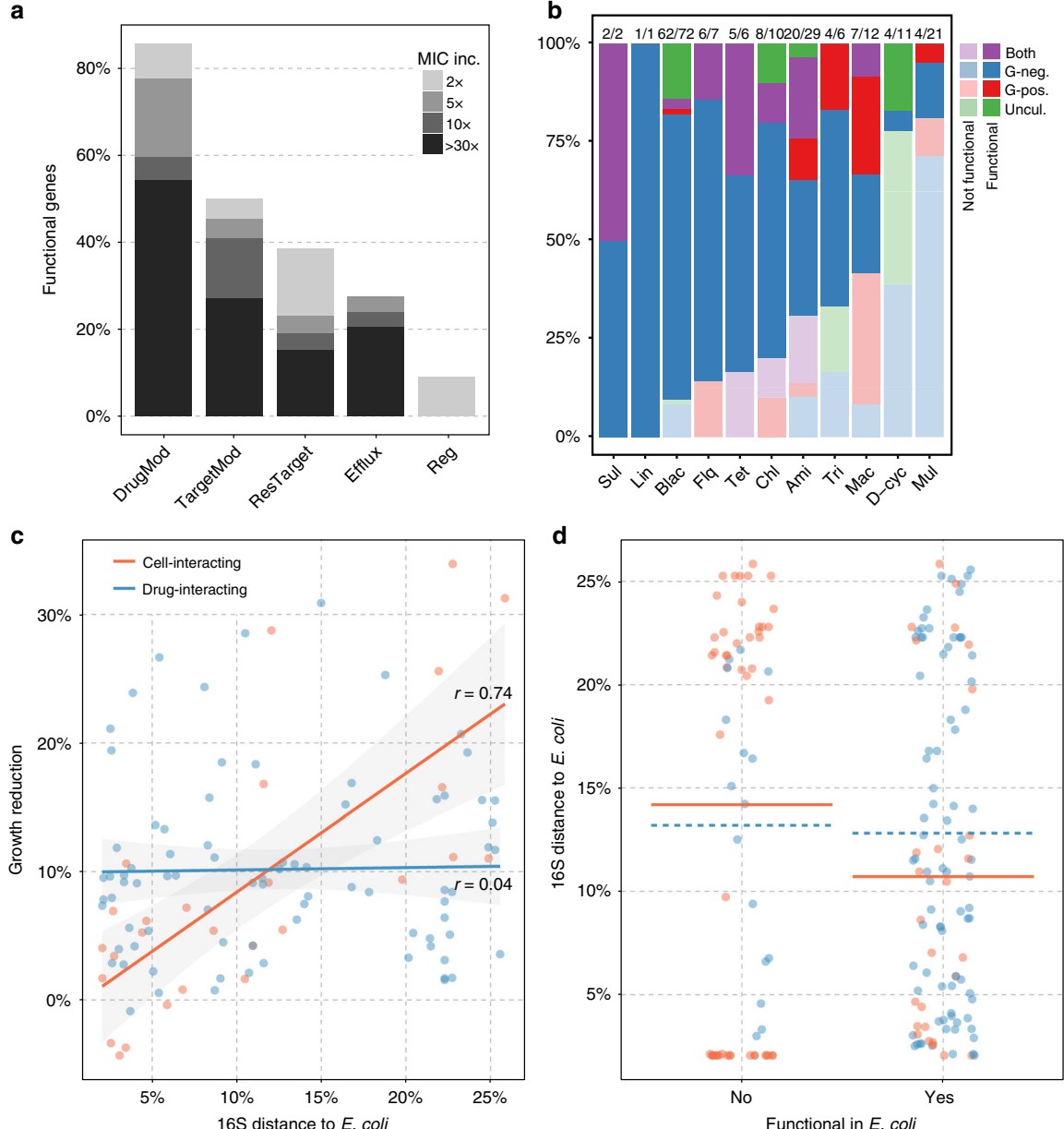

**Fig. 5** Distribution of functional genes on phylogenetic affiliation within drug classes and resistance mechanism. **a** Functionality and resistance level of the resistance genes belonging to the different mechanistic categories. DrugMod: drug inactivating enzymes (n = 113), TargetMod: target modifying/binding proteins (n = 20), ResTarget: redundant/resistant target (n = 29), Efflux: efflux pumps (n = 28), Reg: regulators (n = 10). **b** The number of functional genes annotated as conferring resistance towards each drug class are colored according to their affiliations with Gram-negative organisms (G-neg.); Gram-positive organisms (G-pos.); both Gram-negative and Gram-positive organisms (both); or none of currently sequenced genomes in RefSeq (Uncul.). The number of functional genes out of the total genes within each class is indicated above the bars. **c** The average 16S rRNA distance between E. coli and all RefSeq genomes where each gene has been identified correlated with the growth reduction inflicted by each gene. A linear regression model was fitted to the subsets of genes interacting with the drug (n = 74) and cellular components (n = 27), respectively, were phylogenetic data was available. Cell-interacting mediators include those that conferred resistance via target protection or modification, provision of a resistant target, efflux or regulatory interactions. Drug-interacting genes conferred resistance through modification or breakdown of the antibiotic without interfering directly with host physiology. Shades represent the standard error of the linear fit. **d** The phylogenetic distance from E. coli for functional and non-functional genes tested in E. coli. The mean is shown for the subsets of drug-modifying (blue dashed line) and cell-interacting (orange solid line) resistance mediators

resistance mechanism and genomic relatedness for gene functional compatibility.

## Discussion

To better understand the factors underlying functional compatibility and the fitness cost of genes in new hosts, we assessed a set of diverse antibiotic resistance genes spanning a wide range of

sequence compositions and biochemical mechanisms. We found that a substantial proportion of genes, not previously observed in E. coli, were functional despite huge sequence deviations, in the CAI and GC content, from the E. coli genome average. Interestingly, we found that sequence composition, including the CAI, N-terminal mRNA-folding energy, and GC content, was a poor predictor of functional compatibility and fitness cost (Figs. 3 and 4). Previous studies have shown that the expression level of

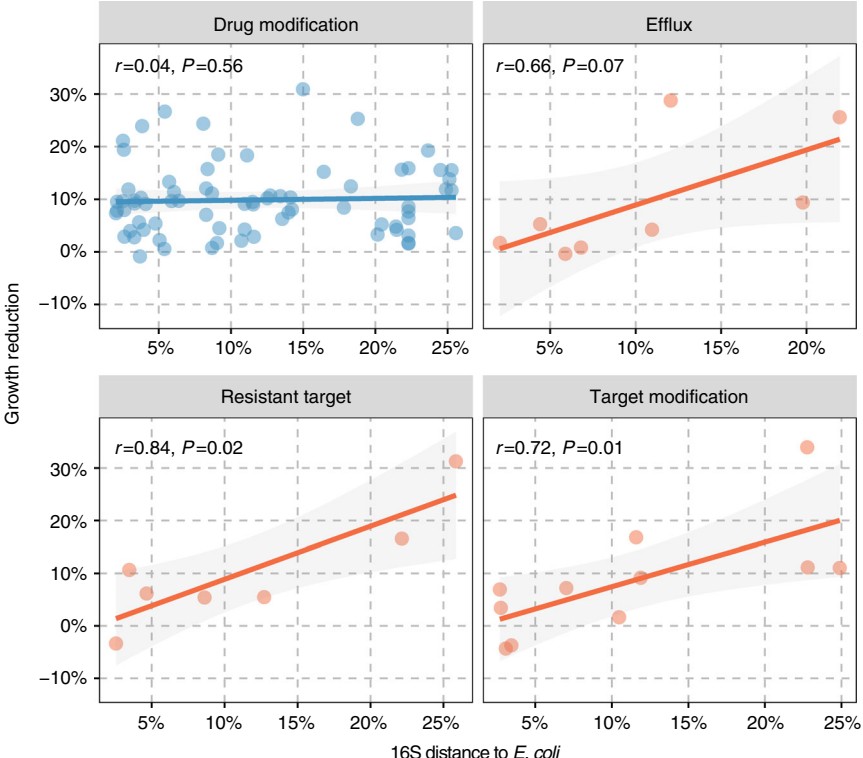

**Fig. 6** The correlation of the average 16S distance with the growth reduction imposed by genes of different mechanistic categories. There was no significant difference, in the observed correlations, between the cell-interacting categories (orange) in a multiple linear regression (ANOVA, $P = 0.93$). Regulators were excluded because this mechanistic category only contained one functional gene (*cpxA*). Shading represents the standard error of the linear fit and the Spearman correlation coefficients are shown

codon-randomized *gfp* variants is mainly affected by the N-terminal mRNA-folding energy; however, strong negative effects of mRNA folding are expected to be counter selected in naturally occurring genes, and we found no significant correlation with this parameter and functionality in our diverse gene set[26,39]. Although we did not assess the effect of sequence composition on expression levels directly, our results suggest that gene expression is not trivially linked to phenotypic output as is the case for fluorescent proteins (Supplementary Fig. 2)[26]. By virtue of resistance phenotypes, we were able to measure end-point functionality directly to avoid indirect, e.g., protein-level, measurements that might not reflect correct phenotypic integration.

Using antibiotic resistance genes as a model for transferable accessory genes, we experimentally showed that the mechanism of the gene product is more important for its functional compatibility than the gene sequence composition (Fig. 5a). The results obtained for the antibiotic resistance genes included here are likely to apply more broadly to horizontal gene transfer, and our experimental results support the "complexity hypothesis" originally proposed by Jain et al., which state that highly interactive gene products are less likely to undergo horizontal transfer[9,41,42]. Although we did not assess protein interactions directly, we observed that genes encoding regulators and efflux pumps, which are dependent on regulatory networks or cell envelope structures, were the least likely to function when expressed in *E. coli* compared to genes encoding enzymes that act directly on the drug (Fig. 5a). This result signifies that the extent of physiological decontextualization dictates the likelihood of selection following gene transfer.

While the trend towards higher costs of target-modifying mediators conferring high-level resistance might be expected, the lack of, and even opposite tendencies, observed for the remaining

mechanistic classes, suggests that high resistance and low fitness cost might not be opposing features (Supplementary Fig. 6).

The biological cost of functionally expressing a new gene is an important factor that potentially determines the reversal of antibiotic resistance upon cessation of antibiotic use[22]. Fitness effects are believed to affect the long-term success of gene transfer events as well as the robustness of engineered biological systems, and the origins of these costs have been suggested to stem from suboptimal sequence composition[18,43,44]. Notably, in our setup of moderate expression, our growth measurements did not detect a significant influence of the sequence-level parameters previously suggested to influence the growth rate of *E. coli* expressing heterologous proteins[26,27]. These studies employed libraries of lower or similar sequence-level diversity compared to our data set, albeit with a narrow mechanistic focus (GFP and ϕ29 DNAP), and observed that the CAI and GC content affected the growth rate of *E. coli*[26,27]. Yet, in line with our observations, Knöppel and Lind et al.[45] were unable to measure a significant fitness effect of parameters such as GC content and length of random DNA inserts, containing a variable number of open reading frames of unknown functions, when expressed from a single copy on the chromosome[45]. However, we acknowledge that these effects exist and may be measured by more sensitive fitness assays or at higher expression levels than the ones used here. Compared to the linear influence on the growth rate of *E. coli* expressing *gfp* and ϕ29 DNA polymerase genes within the narrow GC range (40.4–53.7%) observed by Raghavan et al.[27], our data suggest that the possible effect of GC content is non-linear, and that the deviation from the host genome is more important than the absolute GC content (Fig. 4b and Supplementary Fig. 8).

While previous studies have found a correlation between fitness and evolutionary distance for homologs of *E. coli* genes

involved in translation[29,30], we show that these effects are also evident for accessory genes with non-essential and diverse functionalities, which may contribute to the phylogenetic barriers more generally observed for horizontal gene transfer[1,13,46]. These fitness effects correlated with the relatedness of the donor and recipient species and were independent of sub-mechanistic- and phylogenetic categories (Fig. 6 and Supplementary Fig. 11). Interestingly, although phylogenetic distance was a much stronger predictor of growth influence than the deviation in GC content, the correlation between GC deviation and phylogenetic distance might explain the limited deviation in GC content observed for transferred genes and their recipient genomes (Supplementary Fig. 9)[15]. However, by demonstrating the dependence on cellular interactions, and showing that GC optimization of the *dfrG* gene did not improve its cost, GC content may be a confounder rather than the cause of the fitness effects observed here.

We found that sequence-level properties have a limited impact on heterologous gene fitness and function and are therefore unlikely to be the dominant causal factors confining genes within specific phylogenies. Instead, we show that biochemical mechanisms have strong impacts on heterologous gene fitness that are proportional to the phylogenetic distance to its customary host. The notion that the functional compatibility of cell-interacting proteins is dependent on phylogenetic relationships supports a shift in focus away from sequence composition and metabolic constraints as limiting factors in horizontal gene transfer. Interestingly, the positive effects observed for certain members of the *dfr* and *qnr* families, implicated in DNA gyrase protection and folate metabolism, might even enhance their persistence in the absence of antibiotic selection. However, this is only true for gene variants acquired from closely related species and these fitness effects likely depend on the environmental conditions; e.g., the growth medium.

Importantly, our data suggested that antibiotic resistance genes interacting directly with the drug are more likely to function and be maintained when transferred to a new host. Historically, drug-interacting resistance mechanisms, e.g., utilized by aminoglycosides, chloramphenicol and β-lactam resistance genes, have emerged faster following the introduction of the drug class in question, compared to, e.g., macrolide and vancomycin resistance, which are mediated largely through modification of the cellular targets[47]. For aminoglycosides, drug-modifying genes were detected regularly since the late 1940s, whereas ribosome-modifying methyltransferases were first detected in the early 2000s[48].

We believe that the detailed phenotypic information on individual resistance genes obtained here will be an important resource for ranking the risk of resistance genes and predicting their evolution against existing and future drugs[34,49]. However, knowledge on antibiotic use, co-selection, regulatory, or compensatory interactions in a range of hosts and growth conditions is needed for more accurate predictions[50]. Coupling our findings to factors such as drug usage and ecology of pathogens may allow the construction of predictive models to help guide rational drug usage and development of novel drug classes that are less susceptible to resistance development[34]. Finally, the concepts derived in this study may also guide metabolic engineering or synthetic biology efforts when heterologous proteins are needed to engineer any kind of robust biological system[5,6].

## Methods

**Database construction and gene synthesis**. All resistance gene entries from the ARDB, CARD and Lahey Clinic β-lactamase databases (accessed December 2014) and functional selection studies were downloaded[36,51–54]. A total of 4253 unique sequences were obtained after clustering at 99% nucleotide identity using CD-HIT

software[55], and these sequences were further clustered at 80% identity resulting in 839 gene clusters. These clusters were first sorted according to the average number of BLAST hits obtained in NCBI to include the most abundant genes and then sorted on cluster size to limit the inclusion of housekeeping genes; that are often conserved and occur in small clusters. The 200 largest clusters were selected based on the modal sequence length of each cluster. Trimming was performed using GeneMarkS to only include the longest open reading frame of each sequence[56]. Due to synthesis and cloning limitations, genes longer than 1970 bp were excluded, and *XbaI* and *AscI* sites were synonymously removed by manual sequence inspection. Two primer binding sites for amplification of the entire gene and a unique barcode were added to each of the 200 sequences (Supplementary Fig. 2). Finally, the sequences were ordered as gBlocks through Integrated DNA Technologies (IDT, Coralville, Iowa, USA). All genes and the data obtained for each functional gene are available in Supplementary Data 1.

**Gene dissemination and genomic context analysis**. To assess gene association with sequenced genomes, BLAST comparisons were performed at a 95% identity alignment cut-off, and validated through EMBOSS Matcher pairwise alignment, against NCBIs RefSeq database (67,704 entries; last performed October 2016). The rarefaction analysis of nucleotide databases (Supplementary Fig. 1b) was performed at a 90% identity and 90% coverage cut-off against the NCBI nucleotide and genomes databases. To quantify the dissemination distance of those antibiotic resistance genes with respect to *E. coli*, we chose one representative 16S rRNA from each genus carrying any ARG in RefSeq. The host distance was then calculated using EMBOSS Matcher pairwise alignment between *E. coli* 16S rRNA and representative 16S rRNA for the ARG-carrying genomes and reported as the percent mismatches in the total alignment of the *E. coli* reference 16S rRNA gene.

**Cloning and expression of gene library**. The pZAT vector backbone (Supplementary Fig. 2) was derived from the medium-copy p15A-based pZA21 vector[57] by exchanging the resistance marker with the *Sh ble* Zeocin resistance gene (Thermo Fisher Scientific, Waltham, MA, USA) and inserting the low/medium strength BBa_J23110 promoter of the iGEM parts registry (http://parts.igem.org). This backbone was PCR amplified using primers 5′-AATTTGGCGCGCCCATCAAA-TAAAACGAAAGGC-3′ and 5′-AATTTTCTAGATCTCCTCTTTAATGCTCGC-3′ to create *XbaI* and *AscI* digestible overhangs. The PCR product (25 μL) was subsequently treated with 0.5 μL DpnI restriction enzyme (Thermo Fisher) overnight (O/N) at 37 °C to remove residual template and subsequently PCR purified (NucleoSpin Gel and PCR Clean-up kit from Macherey-Nagel, standard protocol). In 200 individual reactions, the vector backbone was combined with each synthetic gene block in a reaction containing *XbaI* and *AscI* restriction enzymes (Thermo Fisher). The reactions were incubated at 37 °C for 1 h, followed by heat inactivation of the enzymes at 80 °C for 20 min. Subsequently, the 20 μL reactions were ligated by adding 0.5 μL T4 ligase and 3 μL ATP (5 mM) and incubated at room temperature O/N. The 5 μL ligation product was used to transform 50 μL *E. coli* MG1655 (the same strain previously used and sequenced in Munck et al.[58]) chemically competent cells followed by recovery at 37 °C for 3 h. Transformed cells were selected on Zeocin-containing plates (40 μg/ml), and correct insertion of gene blocks was verified by colony PCR and subsequent Sanger sequencing using primers 5′-TATGCCGATATACTATGC-3′ and 5′-AAGCACTTCACTGACACC-3′.

**Antibiotic susceptibility testing**. The minimal inhibitory concentration of all 20 included antibiotics was measured for all library clones as well as *E. coli* MG1655 carrying the empty vector backbone (pZAT). One colony of each clone was inoculated in 180 μL LB/well (with 40 μg/ml Zeocin added for backbone selection) for overnight (O/N) pre-culturing at 37 °C. Ninety-six-well plates were prepared with 100 μL MHB2 medium (Sigma) per well, containing the respective antibiotics, at concentrations 2×, 5×, 10×, and 30× the MIC of *E. coli* MG 1655/pZAT (Supplementary Table 1). Three replicate plates were inoculated with 5·10⁵ cells and incubated for 18 h at 37 °C with shaking at 250 rpm (Titramax 1000, Heidolph). Endpoint optical density (OD) was measured at 600 nms (Synergy H1, BioTek), and the MIC was defined as the highest concentration with less or similar absorbance as the *E. coli* MG1655/pZAT (negative control) subjected to the same antibiotic concentration.

**Growth rate measurements**. The growth rate of each functional clone was measured as the maximum increase in OD over time during exponential growth. Individual colonies were picked and placed onto a pre-inoculation plate and grown for 2–3 h with shaking at 37 °C before inoculation of the final measurement plate. Breathe-Easy (Sigma-Aldrich) film was applied to minimize evaporation during measurements. OD measurements were conducted in 96-well plates containing 150 μL LB medium per well by the ELx808 plate reader (BioTek, USA). OD at 600 nms was measured over 5-min intervals for a maximum of 16 h, and the plates were incubated at the medium shaking setting at 37 °C between measurements.

**Sequence parameters and statistical analysis**. All statistical analyses were performed using R (version 3.1.1). Sequence composition data were obtained using the native functions of Biopython (version 1.70)[59]. The mRNA-folding energy was calculated for 35 nt up- and downstream the start codon of each gene using the

*RNAfold* web server (http://rna.tbi.univie.ac.at/). The non-parametric Mann–Whitney *U* test was used to compare sample means, and a $\chi^2$ test was used when frequencies were assessed. Spearman's rank correlations were performed to assess the strength of associations between two continuous variables. When the influence of multiple variables was assessed simultaneously, a generalized linear model was fitted to binary response variables (*glm* function in R), and a multiple linear regression was fitted (*lm* and *anova* functions in R) to continuous response data.

**Data availability**. The authors declare that all the relevant data are provided in this published article and its Supplementary Information files, or are available from the corresponding author on request.

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

## Acknowledgements

We thank Peter Rugbjerg and Leonie Jahn for critical reading and suggestions to the manuscript. M.O.A.S. further acknowledges financial support from the Novo Nordisk Foundation, the Lundbeck Foundation and the Danish Council for Independent Research.

## Author contributions

A.P., M.O.A.S., and C.M. conceived the study. T.S.S. and A.P. performed the experimental work. A.P., T.S.S., and C.M. analyzed the data. A.P., C.M., and M.M.H.E. compiled the antibiotic resistance gene database, and M.M.H.E. provided genomic association data. A.P. and M.O.A.S. wrote the manuscript with input from C.M., T.S.S., and M.M.H.E.

## Additional information

**Competing interests:** The authors declare no competing financial interests.

