## [Peer Review File · Nature Communications]

Reviewers' comments:

Reviewer #1 (Remarks to the Author):

In this manuscript, Porse et al experimentally examined 200 antibiotic resistance genes and tried to assess the functional compatibility of these horizontally transferred genes in *E. coli*. They argued that the phylogenetic origin and biochemical interactions are the major factors governing the functionality and fitness of resistance genes. The functional compatibility of horizontally transferred genes is an important research topic and research like in this manuscript is potentially promising for the understanding of functional consequences of horizontal gene transfer. Unfortunately, this manuscript in the current form suffers missing key figures and poor presentation. The manuscript would benefit from some well-controlled analyses to directly support their main statements.

Major issues:

1. The main conclusion of the manuscript is that "the phylogenetic origin and the extent to which a gene interacts biochemically with host physiology, are major factors governing the functionality and fitness of resistance genes". There were no real data in the manuscript supporting the biochemical interactions (Supplementary Figure 9 was cited in the manuscript, but Supplementary Figures 9-10 are absent in the manuscript). The main Figure 5 contains some data to illustrate the association with phylogenetic distance. But no efforts were made to address whether (or how much of) the observed trends are due to the biased (phylogenetic and functional) distribution of different resistance genes.

2. 68% of the tested genes from their data set have not been identified in *E. coli* (line 142), and a majority of the genes are from gamma-proteobacteria, following by beta-proteobacteria, Actinobacteria and bacilli etc. In addition, genes from specific clades show noticeable differences (or biases) in classes. In other to claim that phylogenetic origin and biochemical interactions are important in the functional compatibility of transferred genes. It would be essential to make efforts to tease them apart. It would be possible to conduct analyses on the genes that have been identified in *E. coli*. It would also be possible to conduct separate analyses on genes from gamma-proteobacteria. It would also be possible to conduct separate analyses based on gene types. Statements made without such a control would seem unconvincing. To robustly demonstrate what they want to demonstrate, the authors may want to consider methods such as the phylogenetic independent contrasts.

3. Growth reduction was used as an indicator for fitness. The authors should be aware that growth reduction is insensitive. It is nice to know that correlation was found even using growth reduction. But it is very dangerous to reject other hypotheses based on insufficient data. If the authors were to use some more sensitive measure, e.g., competition assay, together with some well controlled experimental design, the authors would likely be able to address the contribution of different gene properties to the functional compatibility of horizontally transferred genes.

Minor comments:

4. The introduction is very narrow and relied on just a few review papers. I would suggest the authors to widen their view and maybe even add some text on the evolutionary process of horizontal gene transfer, which can actually benefit the presentation on functional aspects of HGT.

5. The genes were grouped into different classes and types. Are there any overlaps?

6. The groups and classes in Figure 1 should be better ordered and organized. How the groups in a and b are relevant to one another. The description on the abbreviations in the Figure legend are not even in the same order as in the Figure.

7. The authors used both parametric and non-parametric statistical tests in the study without any justification. Given the uncertainty of data distribution (or likely non-normal distribution shown in their Figures), it would be more appropriate to use non-parametric tests throughout their study.

8. The authors used linear regression to evaluate the correlation associated with the 16S rRNA. It is nice to know correlation was observed. They should be aware that 16S rRNA is not the best model for molecular clock study in bacteria. For molecular clock purpose, the conserved protein genes may be better alternatives. As mentioned in 7, correlation should be best tested using non-parametric methods.

Reviewer #2 (Remarks to the Author):

This manuscript describes a large-scale analysis of 200 resistance genes transformed into *E. coli*, and explores how they affect fitness in the presence or absence of the drug. Analysis of this scale is of considerable interest to the antibiotic resistance field and also to HGT research. The experimental work and the statistic are well-described and appear well-performed, and the manuscript is clearly written and easy to follow. However, previous studies directly relevant to this work are often either completely ignored or referred to only superficially. This results in two major problems. The first exaggerated novelty is expected, but really not required – the sheer scale of this project makes it beyond comparison to its predecessors. The second, which is more serious, has to do with providing the right context for interpretation of the findings of this work. I have detailed below specific places in the introduction and discussion that require attention. There are also a few grammatical errors, typos and missing spaces that can be taken care of by the English editors of the journal.

Introduction

There have been several studies simulating gene transfer of a horizontally acquired resistance genes encoded with different synonymous versions, most notably the one by the Bravo group (*Mol Biol Evol.* 2010 Sep;27(9):2141-51). It is surprising that several genomics-based studies such as those by Puigbo and collaborators from the Koonin lab (*Genome Biol Evol.* 2010; 2: 745–756.) are not mentioned since they nicely show which functions are more frequently transferred, including simple metabolic functions and resistance.

Discussion

"Previous studies have shown that expression level is mainly affected by the N-terminal mRNA-folding energy; however, we found no significant correlation with this parameter and functionality in our diverse gene set" – Here the authors inaccurately cite a paper showing that highly-expressed genes are rarely transferred, which should be cited when discussing the fitness cost of expressing a foreign (otherwise neutral) gene. More importantly they cite the famous Kudla paper that their data seem to contradict. This comparison is somewhat naïve if not downright misleading. First, using synthetic genes with semi random composition, as Kudla did, shows the large effects of mRNA folding, but in biological systems strong mRNA structures at the N-terminal, that prevent or reduce translation too strongly, have been selected against. Thus, if one looks at *E. coli* data protein abundance is better correlated with CAI than mRNA folding (see for example Fig. 2 of *Proc Natl Acad Sci U S A.* 2010 Feb 23; 107(8): 3645–3650.). The authors used natural coding genes from other organisms, so their findings re mRNA should be put in the right context. As for the cost of expression of genes encoded by rare codons, low CAI etc, such costs are only observed under high expression, as in Kudla et al, and the low copy vector used here with a low/medium strength promoter should not result in such costs (again see work from Bravo and collaborators).

"we observed that genes encoding regulator and efflux pumps, which are dependent on regulatory

networks or cell envelope structures, were the least likely to function when expressed in *E. coli* compared to genes encoding enzymes
least likely to function when expressed in *E. coli* compared to genes encoding enzymes that directly on the drug"

This is an important although somewhat expected finding. I would expect in this case a test that goes beyond the 16S relationship and tests whether nearly all Gram-negative pumps that work with a double membrane structure and nearly all regulators from gamma-proteobacteria work since regulatory sequences tend to be similar.

When discussing fitness cost of expression in genes with different GC, the comparison with Raghavan is important but the differences between the studies are profound, and again understated. That study looked at just two genes (though not one as the authors imply), with different codons encoding them, and showed higher fitness with high GC whereas here there are much more genes over a broad GC spectrum but they do not encode the same protein sequence. Nevertheless, the authors are completely correct in concluding that based on their data deviation from GC is more important than absolute.

" However, by demonstrating the dependence on cellular interactions and showing that GC optimization of the *dfrG* gene did not improve its cost, GC-content may be a confounder rather than the cause of the fitness effects observed" – although I agree that GC-content is most likely a confounder and not a cause a single gene optimization experiment (anecdotal, though nice) is not really evidence that strongly supports such a claim.

Reviewer #3 (Remarks to the Author):

As stated in lines 64-68, the authors pointed out that resistance mechanisms and the phylogenetic relatedness of donor and recipient species are critical to determine foreign gene functionality (here focused on antibiotic resistance) and fitness cost, more than sequence composition. Even if the authors claim that they "discover" this issue -and indeed they significantly pushed up the hypothesis in this manuscript- there are illustrious precedents, of course Jain and Rivera, and Andersson and Hughes, rightly mentioned in the text, but also others. In any case, this manuscript has enough merits to be considered, after revision, as publishable in our Journal.

Line 2: Title.

Biochemical mechanism limits the functional compatibility of heterologous genes

Why not "Biochemical mechanisms?"

Why not to say "... functional compatibility of heterogeneous antibiotic resistance genes", as the work essentially focus on these genes, not on heterogeneous genes in general.

See also (at the end of this text) comments on lines 346-340.

Lines 73-78, and 95-107

The first part of the "Results" section is in fact part of the "Material and Methods". The authors describe the species content and the gene cluster structure of the studied sample, but that is not a result concerning the biological reality, as is fully dependent on the known sampling biases of resistance gene databases, with hyper-representation of some species and clones.

Lines 101-107

The authors should clearly state that there are focusing antibiotic-resistance genes, including those for unknown drug targets. Maybe what they propose is applicable to other genes acquired by horizontal gene transfer, and that should be mentioned and discussed. In the current text this set of "accessory genes" is not duly mentioned. In fact, most of the authors' exercise incardinate in

the comparison between the phylogenies of core and accessory genes. Once the accessory set of genes is identifiable, bioinformatic comparisons among core-accessory phylogenies is possible (Accessory Genome Constellation Network): comparative genomics software for accessory genome analysis using bipartite networks. Val F. Lanza et al., 2017. *Bioinformatics* 33:283–285, <https://doi.org/10.1093/bioinformatics/btw601>), with conclusions in the same line of those obtained in the present work.

The question of “What is a resistance gene?” is also critical in the context of this manuscript. In the publication of Martínez, J. L. et al (2015). What is a resistance gene? Ranking risk in resistomes. *Nature Reviews Microbiology*, 13(2), 116-123, the question of the burden imposed by the “connectivity” as a requirement for functionality of resistance determinants with other genes is mentioned, quoting previous hypothesis.

Lines 115-125.

It should be clear for the reader that, as *E. coli* was used as the only recipient of 200 selected genes, those genes expressing some resistance should be preferentially among those that have been detected in *E. coli* or closely related organisms (also here there is an obvious selection bias in databases). In this context, to mention as a “result” that “63 % of the 200 genes displayed at least one resistance phenotype in *E. coli*” says nothing without the information in the same paragraph about the origin of these genes. If the 200 genes were predominantly of Gram-positives, the “result” of 63% will be certainly other. The “results” about different types of antibiotics that are or not functionally expressed also reflects the sampling bias in the gene collection, and the statistical statements (as in lines 124-125): “This difference in resistance level between drug classes was statistically significant (ANOVA, $P < 0.001$)” are difficult to be applicable in this case. We would suggest for the next work to use a more homogeneous set of resistance genes (for instance only using those of Enterobacteriaceae, or Gamma-Proteobacteria). The demonstration of the role of phylogeny could be much better addressed than in the present work. The authors indirectly acknowledge this bias in lines 189-190 when addressing the different functionality in *E. coli* of trimethoprim resistance genes from *E. coli* or *S. aureus*.

Lines 168-180, Fig 2

Even considering the previous remarks, the data about resistance levels and relative growth are certainly of interest. Again, there is a number of “obvious results”, for instance when some drugs are tested on *E. coli*, and this strain is naturally (intrinsically) resistant to them. In Figure 2, some of the acronyms are absent in the legend, as AMX, MEC....

“Contrary to current thinking, some antibiotic resistance genes from the *qnr* and *dfr* families were slightly advantageous to *E. coli* in the absence of antibiotic selection”. The reader understand that the recipient *E. coli* retain for instance the normal set of *dfr* genes; a surplus of these biosynthetic genes results in better growth. Is that the suitable explanation?

Lines 226-233

Again, any type of “statistics” and “percentages” does not provide here an image of the natural “reality”. Of course, “The highest proportion of functional genes was found among the drug-modifying enzymes, including the β -lactamases and aminoglycoside transferases...”, probably as many of these genes have been characterized in *E. coli* or related organisms (a *petitio principii* matter).

However, the general conclusion (hypothesis) that “genes involved in limited cellular interactions are more likely to be functionally compatible in a new host”, c frequency of corroborating the complexity hypothesis (Jain&Rivera, rightly quoted in line 301), and the higher frequency of “de-contextualized genes” (and capture by elements involved in horizontal gene transfer favors de-contextualization, see above comments of lines 101-107), is certainly worth to be highlighted again. De-contextualization means limited metabolic interactions. Of course, drug-detoxifying enzymes, just targeting an “external” molecule, the drug, should those with a lesser effect on host metabolism, and those with lower fitness costs. The authors successfully illustrate this point, educatively differentiating drug-oriented and cell-oriented type of drugs.

Lines 249-258

Phylogenetic distance affects fitness for cell-interacting resistance mechanisms. Of course, "cell-oriented" drugs, influencing metabolism, should reduce fitness. The "evolutionary maturation" of a resistance gene in a particular genomic context tends to decrease such fitness cost. That is well taken, and provides an explanation to the relative absence of effect of sequence composition (line 217).

Line 267-271.

See again our comment of lines 101-107 (above) about the possibility of comparing phylogenies of accessory (including antibiotic resistance genes) and core genomes.

Line 346-340

As we mentioned before (comment on lines 226-233, at the end) the main message is the differentiation of drug-oriented versus cell-oriented drugs (maybe even that merits to be included in the Title) as two pivotal features orientating the evolution of antibiotic resistance.

Reviewers' comments:

Reviewer #1 (Remarks to the Author):

In this manuscript, Porse et al experimentally examined 200 antibiotic resistance genes and tried to assess the functional compatibility of these horizontally transferred genes in *E. coli*. They argued that the phylogenetic origin and biochemical interactions are the major factors governing the functionality and fitness of resistance genes. The functional compatibility of horizontally transferred genes is an important research topic and research like in this manuscript is potentially promising for the understanding of functional consequences of horizontal gene transfer. Unfortunately, this manuscript in the current form suffers missing key figures and poor presentation. The manuscript would benefit from some well-controlled analyses to directly support their main statements.

We are glad the reviewer found our manuscript interesting and we are very thankful for the time spend on the clear and detailed review below. We hope that our response satisfies the reviewer.

Major issues:

1. The main conclusion of the manuscript is that "the phylogenetic origin and the extent to which a gene interacts biochemically with host physiology, are major factors governing the functionality and fitness of resistance genes". There were no real data in the manuscript supporting the biochemical interactions (Supplementary Figure 9 was cited in the manuscript, but Supplementary Figures 9-10 are absent in the manuscript). The main Figure 5 contains some data to illustrate the association with phylogenetic distance. But no efforts were made to address whether (or how much of) the observed trends are due to the biased (phylogenetic and functional) distribution of different resistance genes.

We thank the reviewer for pointing out the overstated sentence in the conclusion of the manuscript. We inferred the physiological interactions of each resistance gene from their functional annotations (obtained via ResFams). These functional categories are well supported in the literature on resistance mechanisms, and we agree with the reviewer that these are merely qualitative (whether or not a resistance mechanism depends on cellular structures for its function or not). As no quantitative data (e.g. protein-protein interaction studies) exist for these genes, we only did analysis within the resolution permitted by this grouping (cell- or drug-interacting).

To underline the qualitative nature of our results, we have rephrased the sentence referred to by the reviewer:

Line 18-20:

"Instead, we identify the phylogenetic origin and the dependence of a resistance mechanism on host physiology as major factors governing the functionality and fitness of antibiotic resistance genes."

We are glad the reviewer noticed the interrupted ordering of our supplementary figures. The numbering was accidentally shifted and this mistake has now been corrected in the manuscript and supplement.

2. 68% of the tested genes from their data set have not been identified in *E. coli* (line 142), and a majority of the genes are from gamma-proteobacteria, following by beta-proteobacteria, Actinobacteria and bacilli etc. In addition, genes from specific clades show noticeable differences (or biases) in classes. In other to claim that phylogenetic origin and biochemical interactions are important in the functional compatibility of transferred genes. It would be essential to make efforts to tease them apart. It would be possible to conduct analyses on the genes that have been identified in *E. coli*. It would also be possible to conduct separate analyses on genes from gamma-proteobacteria. It would also be possible to conduct separate analyses based on gene types. Statements made without such a control would seem unconvincing. To robustly demonstrate what they want to demonstrate, the authors may want to consider methods such as the phylogenetic independent contrasts.

We are very grateful for the excellent suggestions from the reviewer. The reviewer is indeed right, that a deeper analysis of the observed phenomenon will heighten the quality of our manuscript.

We have followed the advice of the reviewer and incorporated extra analysis to clarify the potential role of differences in mechanistic-, drug- and phylogenetic classes in biasing our general observations. By doing so, we see a high degree of concordance across mechanistic-, drug- and phylogenetic categories, and these results have been incorporated into the manuscript and additional supplementary figures (11 and 12).

We have added a short statement to the results; Line 282-284:

*“This correlation was not biased by differences in cell-interacting mechanistic categories (ANOVA, $P = 0.93$; **Supplementary Fig. 11**), targeted drug classes (ANOVA, $P = 0.57$) or phylogenetic groups (ANOVA, $P = 0.23$; **Supplementary Fig. 12**).”*

And we have updated the discussion; Line 359-363:

*“...we show that these effects are also evident for accessory genes with non-essential and diverse functionalities, which may contribute to the phylogenetic barriers observed for horizontal gene transfer¹⁻³. These fitness effects correlated with the relatedness of the donor and recipient species and were independent of sub-mechanistic- and phylogenetic categories (**Supplementary Fig. 11 and 12**).”*

3. Growth reduction was used as an indicator for fitness. The authors should be aware that growth reduction is insensitive. It is nice to know that correlation was found even using growth reduction. But it is very dangerous to reject other hypotheses based on insufficient data. If the authors were to use some more sensitive measure, e.g., competition assay, together with some well controlled experimental design, the authors would likely be able to address the contribution of different gene properties to the functional compatibility of horizontally transferred genes.

We thank the reviewer for bringing up the important aspect of growth rate measurements as a fitness proxy.

We are very aware that the resolution of growth measurements, while indicative, can be too low for distinguishing minor costs. However, the growth rate metric is widely used similar studies showing effects sequence-parameters (E.g. Kudla et al. 2009 and Raghavan et al. 2012) and it generally seems that studies employing both head-to-head competitions and growth measurements observe a good correlation between the two (See Vogwill and Maclean 2015 , DOI: 10.1111/eva.12202).

It was never our intention to reject the hypothesis that sequence parameters can influence fitness, but merely to nuance the picture and highlight additional influential factors that may be relevant for naturally occurring genes at natural expression levels.

We have made sure to phrase our statements as to not reject the hypothesis that sequence-parameters can have (minor) effects on fitness e.g.:

In the abstract Line 16-17: *“GC-content, codon usage and mRNA-folding energy are of minor importance for mechanistically diverse gene products”*

Line 78-79: *“Consequently, we suggest that these effects dominate the potential transfer barriers imposed by sub-optimal sequence composition of heterologous genes.”*

Line 137-139: *“however we could not detect any significant correlations between these sequence parameters and the resistance level for the total dataset nor within resistance classes (Supplementary Fig. 3).”*

Line 344-346: *“Notably, in our setup of moderate expression, our growth measurements did not detect a significant influence of sequence-level parameters previously suggested to influence the growth rate of E. coli expressing heterologous proteins^{20,21”}*

In addition, we have added a sentence to the discussion acknowledging the limited sensitivity of growth rate measurements; Line 352-353:

“However, we acknowledge that these effects exist and may be measured by more sensitive fitness assays or at higher expression levels than the ones used here.”

Minor comments:

4. The introduction is very narrow and relied on just a few review papers. I would suggest the authors to widen their view and maybe even add some text on the evolutionary process of horizontal gene transfer, which can actually benefit the presentation on functional aspects of HGT.

We appreciate the suggestions proposed by the reviewer and we have rewritten and updated the introduction broaden the perspective on HGT and current experimental studies.

On the biased functionality of genes in HGT evolution:

Line 36-41:

“However, the broad-host range of some transfer mechanisms and the ubiquitous presence of antibiotic resistance genes across environments suggest that ecological barriers are largely governed by functional constraints^{7,8}. Indeed, numerous studies reveal a functional bias of transferred gene categories, with e.g. metabolic or virulence factors performing largely independent tasks, being transferred more often than genes encoding highly interactive proteins involved in transcription and translation⁹⁻¹². “

We have added a statement highlighting that evolution of HGT is governed by functionality and fitness:

Line 48-50:

“Upon successful transfer, host compatibility and selection is crucial for the long-term persistence of newly acquired genetic material, as costly genes will eventually be lost through purifying selection¹⁵⁻¹⁸.”

We also added a paragraph summarising recent work on fitness effects of transfer

Line 52-59:

“However, recent work by Kacar and Garmendia et al. showed an increased fitness cost of replacing the elongation factor Tu (a highly conserved protein involved in translation) with its distant homologs in E. coli, and similar results have been obtained by Lind et al. when replacing ribosomal subunits in Salmonella typhimurium^{23,24}. While the fitness cost of expressing these core translational genes could not be attributed to differences in sequence composition, Amorós-Moya et al. showed that sub-optimal codon usage of a highly-expressed chloramphenicol acetyl transferase resistance gene resulted in lower resistance levels and overall host fitness²⁵.”

5. The genes were grouped into different classes and types. Are there any overlaps?

To assist meaningful analysis we grouped the genes into different relevant categories as depicted in the figures and described in the beginning of the results section e.g.:

Line 106-108: *“The selected genes were annotated to confer resistance to 11 distinct drug classes via 23 distinct biochemical functions that can be divided into five major mechanistic categories based on Resfam annotations”*

These overlaps of mechanistic and biochemical categories, and between phylogeny and drug-targeting, are illustrated in Fig. 1, and between Gram-affiliation and drug class in Fig. 5b. All data is available in Supplementary data table 1 (uploaded separately).

To illustrate the overlap further, we have added a new panel to Fig. 1, showing the relation between drug classes and mechanisms (Fig1b).

6. The groups and classes in Figure 1 should be better ordered and organized. How the groups in a and b are relevant to one another. The description on the abbreviations in the Figure legend are not even in the same order as in the Figure.

We thank the reviewer for the suggestions on Figure 1. We have updated Figure 1 to include a panel (b) that connects the drug classes and mechanisms (see also previous comment). The order of the different groups is based on their abundance (a and b) or succession in the stacked bar (c); this is the same order as used in Fig. 2.

We appreciate the suggestion on the order of abbreviations and have changed the figure text accordingly:

Updated Figure 1 text to resemble the order displayed in the legends:

Fig. 1 | Database mechanistic and phylogenetic diversity of 200 synthesized genes. (a) Biochemical functions of the included antibiotic resistance genes. These genes are grouped into five major mechanistic categories (relative abundance shown in parenthesis): DrugMod = Drug inactivating enzymes (56.5%), Efflux = Efflux pumps (14%), Reg = Regulators (5%), ResTarget = Redundant/resistant target (14.5%), TargetMod = Target modifying/binding proteins (10%). (b) Gene counts stratified by resistance class and fractionated by resistance mechanisms. (c) Phylogenetic and drug class distributions of the included genes. Genes that could be identified (97% identity) in one or more genomes deposited in RefSeq were quantified for each bacterial phylogenetic class. The colouring of each bar depicts the distribution of the annotated resistance. Drug class aberrations: Chl = Chloramphenicol, Bla = β -lactams, Ami = Aminoglycosides, Mul = Multiple drug classes, Sul = Sulfamethoxazole, Tet = Tetracyclines, Flq = Fluoroquinolones, Unk = Unknown, Tri = Trimethoprim, Van = Vancomycin Dcy = D-cycloserine, Lin = Lincosamides, Mac = Macrolides.

7. The authors used both parametric and non-parametric statistical tests in the study without any justification. Given the uncertainty of data distribution (or likely non-normal distribution shown in their Figures), it would be more appropriate to use non-parametric tests throughout their study.

We are grateful for the reviewer's suggestion on more robust statistical tests. We generally used the non-parametric Mann-Whitney U-test for all pairwise comparisons throughout the manuscript. We reported one *t*-test statistics for the comparing the growth of codon optimized dfrG variants; however, we have updated this to use the Mann-Whitney U-test for consistency:

We have updated Line 206-207 and Supplementary Fig. 7:

*"...this variant did not show a significant growth improvement compared to the wild-type dfrG gene when expressed in E. coli (Mann-Whitney U-test, $P = 0.1$; **Supplementary Fig. 7**)"*

Similarly, instead of using parametric Pearson correlations, we have now conducted the Spearman's rank correlations where applicable and updated the correlation- and P-values. However, this yielded

largely the same results as obtained with the Pearson correlation, and the overall conclusions are the same.

We have re-written the methods section and updated the figures (4 and 5) accordingly:

Line 471-472 :

"Spearman's rank correlations were performed to assess the strength of associations between two continuous variables."

8. The authors used linear regression to evaluate the correlation associated with the 16S rRNA. It is nice to know correlation was observed. They should be aware that 16S rRNA is not the best model for molecular clock study in bacteria. For molecular clock purpose, the conserved protein genes may be better alternatives. As mentioned in 7, correlation should be best tested using non-parametric methods.

We thank the reviewer for the insightful advice. We chose to use 16S distances because of its familiarity to most readers and to ease comparison with previous studies of barriers in HGT (e.g. Smillie et al. 2011 and Hu et al. 2016) also using 16S distance as a relatedness metric. While there are stronger phylogenetic markers, we believe that the resolution gained is minor compared to other sources of e.g. biological noise in our setup.

Reviewer #2 (Remarks to the Author):

This manuscript describes a large-scale analysis of 200 resistance genes transformed into *E. coli*, and explores how they affect fitness in the presence or absence of the drug. Analysis of this scale is of considerable interest to the antibiotic resistance field and also to HGT research. The experimental work and the statistic are well-described and appear well-performed, and the manuscript is clearly written and easy to follow. However, previous studies directly relevant to this work are often either completely ignored or referred to only superficially. This results in two major problems. The first exaggerated novelty is expected, but really not required – the sheer scale of this project makes it beyond comparison to its predecessors. The second, which is more serious, has to do with **providing the right context for interpretation of the findings of this work**. I have detailed below specific places in the introduction and discussion that require attention. There are also a **few grammatical errors, typos and missing spaces that can be taken care of by the English editors of the journal**.

We are glad the reviewer finds our work interesting. We appreciate that the reviewer has spent time reviewing our manuscript and we thank the reviewer for the insightful suggestions. We have gone carefully through the manuscript to tighten up the grammar and provide answered the reviewer's comments below. We hope that our response is satisfactory.

Introduction

There have been several studies simulating gene transfer of a horizontally acquired resistance genes encoded with different synonymous versions, most notably the one by the Bravo group (**Mol Biol Evol.** 2010 Sep;27(9):2141-51). It is surprising that several genomics-based studies such as those by Puigbo and collaborators from the Koonin lab (**Genome Biol Evol.** 2010; 2: 745–756.) are not mentioned since they nicely show which functions are more frequently transferred, including simple metabolic functions and resistance.

We thank the reviewer for insightful suggestions on additional literature. We agree with the reviewer that our manuscript will benefit from expanding the background on functional biases and experimental gene transfer studies further. We have amended the introduction considerably to include the references suggested by the reviewer and other similar experimental studies on the cost of HGT.

On the biased functionality of transferred genes; referring the study of Pere Puigbo et al. 2010.
Line 36-41:

“However, the broad-host range of some transfer mechanisms, and the ubiquitous presence of antibiotic resistance genes across environments, suggest that ecological barriers are largely governed by functional constraints^{7,8}. Indeed, numerous studies reveal a functional bias of transferred gene categories, with factors performing largely independent tasks e.g. those involved in secondary metabolism or virulence, being transferred more often than genes encoding highly interactive proteins involved in transcription and translation⁹⁻¹².“

Summarising recent work on fitness effects of transfer including Amorós-Moya et al. 2010 (ref 25).
Line 52-59:

“However, recent work by Kacar and Garmendia et al. showed an increased fitness cost of replacing the elongation factor Tu (a highly conserved protein involved in translation) with its distant homologs in E. coli, and similar results have been obtained by Lind et al. when replacing ribosomal subunits in Salmonella typhimurium^{23,24}. While the fitness cost of expressing these core translational genes could not be attributed to differences in sequence composition, Amorós-Moya et al. showed that sub-optimal codon usage of a highly-expressed chloramphenicol acetyl transferase resistance gene resulted in lower resistance levels and overall host fitness²⁵.“

Discussion

"Previous studies have shown that expression level is mainly affected by the N-terminal mRNA-folding energy; however, we found no significant correlation with this parameter and functionality in our diverse gene set" – Here the authors inaccurately cite a paper showing that highly-expressed genes are rarely transferred, which should be cited when discussing the fitness cost of expressing a foreign (otherwise neutral) gene.

We thank the reviewer for pointing out the suboptimal use of the reference by Park et al. We have moved the citation to the discussion of fitness effects on HGT:

Line 341-344:

“Fitness effects are believed to affect the long-term success of gene transfer events as well as the robustness of engineered biological systems, and the origins of these costs have been suggested to stem from suboptimal sequence composition^{22,41,42}”

More importantly they cite the famous Kudla paper that their data seem to contradict. This comparison is somewhat naïve if not downright misleading. First, using synthetic genes with semi random composition, as Kudla did, shows the large effects of mRNA folding, but in biological systems strong mRNA structures at the N-terminal, that prevent or reduce translation too strongly, have been selected against. Thus, if one looks at E. coli data protein abundance is better correlated with CAI than mRNA folding (see for example Fig. 2 of **Proc Natl Acad Sci U S A. 2010 Feb 23; 107(8): 3645–3650.**). The authors used natural coding genes from other organisms, so their findings re mRNA should be put in the right context. As for the cost of expression of genes encoded by rare codons, low CAI etc, such costs are only observed under high expression, as in Kudla et al, and the low copy vector used here with a low/medium strength promoter should not result in such costs (again see work from Bravo and collaborators).

We are grateful for the insightful comments and literature suggestion brought up by the reviewer. The study by Kudla et al. certainly employ a more artificial model library than ours, however, because our library displayed a fair amount of variation in mRNA folding energy (Fig. 2) we decided to include this parameter in our analysis.

After reading the paper by Tuller et al. 2010 we agree completely with the reviewer that the effects of mRNA folding shall not be overstated for natural genes, and we have moderated our statements on the effect of mRNA folding on expression levels accordingly:

In the results section Line 166-167:

“The folding energy of the N-terminal of a transcript may also influence gene expression^{24,26,37}...”

We have updated the discussion to cite the suggested reference and mention the purifying selection against strong mRNA folding in natural genes:

Line 316-320: *“Previous studies have shown that the expression level of codon-randomized gfp variants is mainly affected by the N-terminal mRNA-folding energy; however, strong negative effects of mRNA folding are expected to be counter-selected in naturally occurring genes, and we found no significant correlation with this parameter and functionality in our diverse gene set^{20,41}.”*

In addition, we have updated the discussion to mention the limitations of our setup in detecting potential effects of sequence parameters:

Line 344-346: “Notably, in our setup of moderate expression, our growth measurements did not detect a significant influence of sequence-level parameters previously suggested to influence the growth rate of *E. coli* expressing heterologous proteins^{20,21},”

And Line 352-353:

“However, we acknowledge that these effects exist and may be measured by more sensitive fitness assays or at higher expression levels than the ones used here.”

"we observed that genes encoding regulator and efflux pumps, which are dependent on regulatory networks or cell envelope structures, were the least likely to function when expressed in *E. coli* compared to genes encoding enzymes that directly on the drug" This is an important although somewhat expected finding. I would expect in this case a test that goes beyond the 16S relationship and tests whether nearly all Gram-negative pumps that work with a double membrane structure and nearly all regulators from gamma-proteobacteria work since regulatory sequences tend to be similar.

The reviewer brings up an interesting and relevant point. While there were more Gram-positive genes (2/5) among non-functional efflux pumps compared to functional efflux mediators (1/7) this difference was not significant (χ^2 , $P = 0.083$) and the same was true for regulatory genes at the Gram-class level (χ^2 , $P = 0.38$) and the taxonomical class level (χ^2 , $P = 0.72$).

We have updated the results section:

Line 300-305: “For this subset of genes, the effect of phylogenetic differences was dominated by genes conferring resistance through target replacement (Mann-Whitney U-test, $P = 0.031$) and efflux mechanisms (Mann-Whitney U-test, $P = 0.015$). However, while the Gram-class affiliation was not a significant predictor of functionality for efflux, target replacing and regulatory genes (χ^2 , $P > 0.05$), genes originating from Gram-negatives were significantly overrepresented in functional target modifying resistance mediators (χ^2 , $P > 0.005$). “

When discussing fitness cost of expression in genes with different GC, the comparison with Raghavan is important but the differences between the studies are profound, and again understated. That study looked at just two genes (though not one as the authors imply), with different codons encoding them, and showed higher fitness with high GC whereas here there are much more genes over a broad GC spectrum but they do not encode the same protein sequence. Nevertheless, the authors are completely correct in concluding that based on their data deviation from GC is more important than absolute.

We thank the reviewer again for the observation. While most work on sequence composition and fitness has been conducted using *gfp*, we have amended the text to mention the experiment by Raghavan et al. performed on (the very narrow 44-47% GC range of) sequence variants of the $\phi 29$ DNA polymerase as well.

Line 361-364: *“These studies employed libraries of lower or similar sequence-level diversity compared to our dataset, albeit with a narrow mechanistic focus (GFP and ϕ 29 DNAP), and observed that the CAI and GC-content affected the growth rate of E. coli^{24,25}”*

Line 354-357 *“Compared to the linear influence on the growth rate of E. coli expressing gfp and ϕ 29 DNA polymerase genes within the narrow GC-range (40.4–53.7%) observed by Raghavan et al., our data suggests that the possible effect of GC-content is non-linear, and that the deviation from the host genome is more important than the absolute GC-content”*

"However, by demonstrating the dependence on cellular interactions and showing that GC optimization of the dfrG gene did not improve its cost, GC-content may be a confounder rather than the cause of the fitness effects observed" – although I agree that GC-content is most likely a confounder and not a cause a single gene optimization experiment (anecdotal, though nice) is not really evidence that strongly supports such a claim.

We agree completely with the reviewer. This experiment was merely a test of the GC-hypotheses to show that effects at the protein level were more important in this case. The main argument is supported by the results obtained from our gene library.

Reviewer #3 (Remarks to the Author):

As stated in lines 64-68, the authors pointed out that resistance mechanisms and the phylogenetic relatedness of donor and recipient species are critical to determine foreign gene functionality (here focused on antibiotic resistance) and fitness cost, more than sequence composition. Even if the authors claim that they “discover” this issue -and indeed they significantly pushed up the hypothesis in this manuscript- there are illustrious precedents, of course Jain and Rivera, and Andersson and Hughes, rightly mentioned in the text, but also others. In any case, this manuscript has enough merits to be considered, after revision, as publishable in our Journal.

We thank the reviewer for showing interest in our study and for spending time assessing its quality. We agree that others have hypothesised similarly from *in silico* studies and touched upon the subject experimentally using highly conserved core-genes. However, no experimental work of this scale and diversity with relevance to antibiotic resistance has been done.

Line 2: Title.

Biochemical mechanism limits the functional compatibility of heterologous genes

Why not “Biochemical mechanisms?”

Why not to say “... functional compatibility of heterogeneous antibiotic resistance genes”, as the work essentially focus on these genes, not on heterogeneous genes in general.

See also (at the end of this text) comments on lines 346-340.

We thank the reviewer very much for the suggestion to focus our title. However, we believe that this study represents a diverse set of accessory genes of different mechanic functions (much more than previous studies) and while they all can be assessed via antibiotic resistance phenotypes, we consider them relevant beyond this specific phenotype and are therefore in favour of keeping the title broad.

We have changed “mechanism” to “mechanisms”:

“Biochemical mechanisms limit the functional compatibility of heterologous genes”.

Lines 73-78, and 95-107

The first part of the “Results” section is in fact part of the “Material and Methods”. The authors describe the species content and the gene cluster structure of the studied sample, but that is not a result concerning the biological reality, as is fully dependent on the known sampling biases of resistance gene databases, with hyper-representation of some species and clones.

We are grateful for the reviewer’s suggestion to shorten the results section. We have substantially reduced the introductory paragraph of the results section and refer to materials and methods section instead for details:

Line 82-84: *“By clustering all genes of major publicly available antibiotic resistance gene databases, and selecting the most abundant genotypes, we obtained 200 genes for DNA synthesis (see materials and methods, Supplementary Fig. 1a and Supplementary Table 1).”*

Lines 101-107

The authors should clearly state that there are focusing antibiotic-resistance genes, including those for unknown drug targets. Maybe what they propose is applicable to other genes acquired by horizontal gene transfer, and that should be mentioned and discussed. In the current text this set of “accessory genes” is not duly mentioned. In fact, most of the authors’ exercise incardinate in the comparison between the phylogenies of core and accessory genes. Once the accessory set of genes is identifiable, bioinformatic comparisons among core-accessory phylogenies is possible (Accessory Genome Constellation Network): comparative genomics software for accessory genome analysis using bipartite networks. Val F. Lanza et al., 2017. *Bioinformatics* 33:283–285, <https://doi.org/10.1093/bioinformatics/btw601>, with conclusions in the same line of those obtained in the present work.

We thank the reviewer for the helpful thoughts on accessory genes and antibiotic resistance. We are very aware of reminding the reader that, while these findings are likely to be broadly relevant in HGT, we are focusing on antibiotic resistance phenotypes. We have amended the manuscript to make this point more clear:

Abstract, 13-14: *“...we sampled 200 diverse genes to represent >80% of sequenced antibiotic resistance genes...”*

Line 22-23: *“Instead, we identify the phylogenetic origin, and the dependence of a resistance mechanism on host physiology, as major factors governing the functionality and fitness of antibiotic resistance genes”*

Line 70-73: *“The diversity of mechanisms through which antibiotic resistance is achieved makes antibiotic resistance genes a valuable model system for investigating the factors that may affect the functional compatibility of transferable genes in general”*

Line 108-111: *“In addition to genes annotated to confer resistance towards known drug classes, genes annotated to confer antibiotic resistance but without defined antibiotic resistance profiles were included as a consequence of their high abundance in the public antibiotic resistance databases”*

We have further underlined our gene-set as a model for transferable accessory genes with potential implications for transferred genes in a broader sense, Line 325-329:

“Using antibiotic resistance genes as a model for transferable accessory genes, we experimentally showed that the mechanism of the gene-product is more important for its functional compatibility than the gene sequence composition (Fig. 5a). The results obtained for the antibiotic resistance genes included here are likely to apply more broadly to horizontal gene transfer and our experimental results support the “complexity hypothesis” originally proposed by Jain et al. ...”

The question of “What is a resistance gene”? is also critical in the context of this manuscript. In the publication of Martínez, J. L. et al (2015). What is a resistance gene? Ranking risk in resistomes. Nature Reviews Microbiology, 13(2), 116-123, the question of the burden imposed by the “connectivity” as a requirement for functionality of resistance determinants with other genes is mentioned, quoting previous hypothesis.

The review mentioned by the reviewer is indeed relevant for our work and we have cited the reference accordingly, Line 387-389: *“The detailed phenotypic information on individual resistance genes obtained here will be an important resource for ranking the risk of resistance genes and predicting their evolution against existing and future drugs^{32,47}”*.

Lines 115-125.

It should be clear for the reader that, as E. coli was used as the only recipient of 200 selected genes, those genes expressing some resistance should be preferentially among those that have been detected in E. coli or closely related organisms (also here there is an obvious selection bias in databases). In this context, to mention as a “result” that “63 % of the 200 genes displayed at least one resistance phenotype in E. coli” says nothing without the information in the same paragraph about the origin of these genes. If the 200 genes were predominantly of Gram-positives, the “result” of 63% will be certainly other.

We are very grateful for the reviewer’s suggestion to contextualize our findings and added information on prevalence in E. coli:

Line 124-125: *“Whereas only 32% of the 200 tested genes could be identified in E. coli genomes deposited in NCBI’s RefSeq database, 63 % of the genes displayed at least one resistance phenotype in E. coli”*

The “results” about different types of antibiotics that are or not functionally expressed also reflects the sampling bias in the gene collection, and the statistical statements (as in lines 124-125): “This difference in resistance level between drug classes was statistically significant (ANOVA, $P < 0.001$)” are difficult to be applicable in this case.

We thank the reviewer for mentioning this and we have added an extra panel to Fig. 1, showing the overall abundance of genes targeting different drug classes in among all 200 genes.

We would suggest for the next work to use a more homogeneous set of resistance genes (for instance only using those of Enterobacteriaceae, or Gamma-Proteobacteria). The demonstration of the role of phylogeny could be much better addressed than in the present work. The authors indirectly acknowledge this bias in lines 189-190 when addressing the different functionality in E. coli of trimethoprim resistance genes from E. coli or S. aureus.

We appreciate the suggestion by the reviewer, and we agree that a more rationally compiled database could have heightened the functional aspects of our work. We will definitely consider this for future studies.

Lines 168-180, Fig 2

Even considering the previous remarks, the data about resistance levels and relative growth are certainly of interest. Again, there is a number of “obvious results”, for instance when some drugs are tested on *E. coli*, and this strain is naturally (intrinsically) resistant to them. In Figure 2, some of the acronyms are absent in the legend, as AMX, MEC.....

We are glad the reviewer found our results on resistance level and growth rates interesting. We have detailed our analysis (updated supplementary figure 6) slightly to enhance the mechanistic aspects and trade-off discussions in this regard:

Line 194-197: *“Despite a tendency towards a trade-off between resistance level and growth rate for target-modifying ($r = 0.62$, $P = 0.04$), this trend was opposite for resistant target ($r = -0.61$, $P = 0.06$) and efflux mediators ($r = -0.59$, $P = 0.12$), and no correlation was observed between growth reduction and resistance level for the drug modifying mechanistic class ($r = 0.02$, $P = 0.84$; **Supplementary Fig. 6).**”*

And this has been discussed a bit further at Line 335-338: *“While the trend towards higher costs of target modifying mediators conferring high-level resistance might be expected, the lack of, and even opposite tendencies observed for the remaining mechanistic classes, suggests that high resistance and low fitness cost might not be opposing features (**Supplementary Fig. 6).**”*

We thank the reviewer for noticing the missing abbreviations of in the text of Fig. 2. This has been corrected:

*“The drug classes are as follows: Chl = Chloramphenicol, Bla = β -lactams (Amx = Amoxicillin, Mec = Mecillinam, Cfu = Cefotaxime, Azt = Aztreonam, Mep = Meropenem), Ami = Aminoglycosides, Mul = Multiple drug classes, Sul = Sulfamethoxazole, Tet = Tetracyclines, Flq = Fluoroquinolones, Tri = Trimethoprim, Dcy = D-cycloserine, Lin = Lincosamides, Mac = Macrolides, Col = Colistin, Fos = Fosfomycin, Nit = Nitrofurantoin, Van = Vancomycin. Grey coloured drugs were tested, but no genes conferred resistance towards these in *E. coli*.”*

We agree with the reviewer that the intrinsic resistance of *E. coli* makes some genes less relevant, at least from a clinical perspective, than others. However, because resistance cut-offs can still be obtained for these drugs, and because some (e.g. macrolide) resistance genes are still present in, and transferred between, *E. coli* and similar Gram-negatives, we included these drugs to be comprehensive.

*“Contrary to current thinking, some antibiotic resistance genes from the qnr and dfr families were slightly advantageous to *E. coli* in the absence of antibiotic selection”. The reader understand that the recipient *E. coli* retain for instance the normal set of dfr genes; a surplus of these biosynthetic genes results in better growth. Is that the suitable explanation?*

The reviewer has indeed understood it right and we thank the reviewer for proposing a reasonable explanation for the positive effects of dfr gene expression. We plan to investigate mechanistic basis for the fitness effect of these genes further in a future study.

We have updated the discussion to underline this further.

Line 377-379: *“Interestingly, the positive effects observed for certain members of the dfr and qnr families, implicated in DNA gyrase protection and folate metabolism, might even enhance their persistence in the absence of antibiotic selection, when exchanged between closely related species.”*

Lines 226-233

Again, any type of “statistics” and “percentages” does not provide here an image of the natural “reality”. Of course, “The highest proportion of functional genes was found among the drug-modifying enzymes, including the β -lactamases and aminoglycoside transferases...”, probably as many of these genes have been characterized in E. coli or related organisms (a petitio principii matter).

We thank the reviewer for pointing this out. The reviewer is indeed correct that more drug modifying genes were found in Gram-negatives; however, the Gram-class parameter (and whether they were found in E. coli or not) did not significantly bias the functionality of genes, which we have clarified in the text:

Line 237-241:

“The highest proportion of functional genes was found among the drug-modifying enzymes, including the β -lactamases and aminoglycoside transferases, with most genes conferring high levels of resistance (Fig. 5a), and this distribution was not significantly biased by the phylogenetic affiliation of these genes (χ^2 , $P = 0.08$) or whether they had previously been identified in E. coli (χ^2 , $P = 0.77$)”

However, the general conclusion (hypothesis) that “genes involved in limited cellular interactions are more likely to be functionally compatible in a new host”, a frequency of corroborating the complexity hypothesis (Jain&Rivera, rightly quoted in line 301), and the higher frequency of “de-contextualized genes” (and capture by elements involved in horizontal gene transfer favors de-contextualization, see above comments of lines 101-107), is certainly worth to be highlighted again. De-contextualization means limited metabolic interactions. Of course, drug-detoxifying enzymes, just targeting an “external” molecule, the drug, should those with a lesser effect on host metabolism, and those with lower fitness costs. The authors successfully illustrate this point, educatively differentiating drug-oriented and cell-oriented type of drugs.

We are happy that the reviewer finds our results worthy and our mechanistic categorisation meaningful. We like the decontextualization term used by the reviewer and have highlighted this point further in the discussion:

Line 334-335: *“this result signifies that the extent of physiological decontextualization dictates the likelihood of selection following gene transfer.”*

We have also expanded the introduction of this topic, Line 38-41: *“Indeed, numerous studies reveal a functional bias of transferred gene categories, with factors performing largely independent tasks, e.g. those involved in secondary metabolism or virulence, being transferred more often than genes encoding highly interactive proteins involved in transcription and translation^{9-12”}*

Lines 249-258

Phylogenetic distance affects fitness for cell-interacting resistance mechanisms. Of course, “cell-oriented” drugs, influencing metabolism, should reduce fitness. The “evolutionary maturation” of a resistance gene in a particular genomic context tends to decrease such fitness cost. That is well taken, and provides an explanation to the relative absence of effect of sequence composition (line 217).

We are glad the reviewer finds our conclusions logical and convincing. We are indeed in the process of investigating such “evolutionary maturation” of costly genes.

Line 267-271.

See again our comment of lines 101-107 (above) about the possibility of comparing phylogenies of accessory (including antibiotic resistance genes) and core genomes.

We thank the reviewer again for referring an interesting tool for network analysis, however, we believe that such exhaustive analysis of accessory gene networks in relation to intergenomic relatedness, although very interesting, is beyond the scope of this manuscript and better conducted in a separate study.

Line 346-340

As we mentioned before (comment on lines 226-233, at the end) the main message is the differentiation of drug-oriented versus cell-oriented drugs (maybe even that merits to be included in the Title) as two pivotal features orientating the evolution of antibiotic resistance.

We agree completely with the reviewer that this is the main conclusion of the paper (however, see previous comments on the title). Furthermore, we find the phrasing used by the reviewer so intellectually appealing that we included this statement in the concluding remarks of our abstract:

Line 20-22: *“These findings emphasize the importance of biochemical mechanism for heterologous gene compatibility and suggest physiological constrains as a pivotal feature orienting the evolution of antibiotic resistance.”*

Reviewer #1 (Remarks to the Author):

For the revised manuscript, I am glad to see the addition of Supp Figures 11 and 12. At the same, I noticed that the authors should have done more to interpret these data. For Supp Figure 11, they stated that "There was no significant effect of the different cell-interacting categories (orange) in a multiple linear regression." No information was provided for the P-value and possible explanation in each panel. In Supp Figure 12, there are two types of symbols, but no legend or explanation was provided. P-values and correction coefficients should be provided for each panel. All these should be fixed before it can be published.

In the introduction, "costly genes will eventually be lost through purifying selection 20-23". Citations are all antibiotic resistance genes. To me, they fit in well with the bigger picture that (most) horizontally transferred genes undergo fast rates of turnover. If so, the citations should be several more generalized studies NOT (just) on antibiotic resistance genes. There have been a number of studies on quantifying the processes of gene gain and loss during bacterial genome evolution. The authors might want to cite some of these papers.

The authors observed advantageous growth. It is important to note that advantageous growth is conditional to the specific growth medium (media). The authors should be explicit about this.

Reviewer #3 (Remarks to the Author):

This referee is glad to say that the authors have answered in a satisfactory way to all my queries. I think the manuscript is now much more equilibrated in well discussed.

Reviewer response

For the revised manuscript, I am glad to see the addition of Supp Figures 11 (*now manuscript Fig. 6!*) and 12. At the same, I noticed that the authors should have done more to interpret these data.

We are glad the reviewer liked our improvements and we have added a few lines to summarising the main conclusions from the added analysis:

Line 289-291: *"However, the effects were most pronounced within the target replacing and target modifying mechanistic classes, and for genes affiliated with Proteobacteria and Actinobacteria (Fig. 6 and Supplementary 12)."*

For Supp Figure 11, they stated that "There was no significant effect of the different cell-interacting categories (orange) in a multiple linear regression." No information was provided for the P-value and possible explanation in each panel.

Our intention was to highlight that the correlation does not vary significantly between cell-interacting mechanisms (e.g. is not dominated by one class). We have updated the figure description and added correlation information to each panel:

Fig. 6:

"There was no significant difference, in the observed correlations, between the cell-interacting categories (orange) in a multiple linear regression (ANOVA, $P = 0.93$)."

In Supp Figure 12, there are two types of symbols, but no legend or explanation was provided. P-values and correlation coefficients should be provided for each panel. All these should be fixed before it can be published.

We have added description of the colours in the legend and written correlation information in each panel.

Fig. 6:

"The effect of the average 16S distance on the growth reduction imposed by drug-interacting (blue) and cell-interacting (orange) genes detected in different phylogenetic classes. Only taxonomical classes with more than two genes in each category were included. There was no significant difference in the correlations observed between phylogenetic class affiliations in a multiple linear regression model including these factors (ANOVA, $P = 0.23$)."

In the introduction, "costly genes will eventually be lost through purifying selection 20-23". Citations are all antibiotic resistance genes. To me, they fit in well with the bigger picture that (most) horizontally transferred genes undergo fast rates of turnover. If so, the citations should be several more generalized studies NOT (just) on antibiotic resistance genes. There have been a number of studies on quantifying the processes of gene gain and loss during bacterial genome evolution. The authors might want to cite some of these papers.

We agree with the reviewer that plenty of literature on this phenomenon exist, and we have added additional references to underline the general tendency beyond antibiotic resistance:

Koskiniemi et al., “*Selection-driven gene loss in bacteria*” (2012). PLoS Genetics.

Lee and Marx, “*Repeated, selection-driven genome reduction of accessory genes in experimental populations*” (2012). PLoS Genetics.

Mira et al., “*Deletional bias and the evolution of bacterial genomes*” (2001). Trends in Genetics.

The authors observed advantageous growth. It is important to note that advantageous growth is conditional to the specific growth medium (media). The authors should be explicit about this.

We agree with the reviewer that these effects are likely to be conditional and we have stated this explicitly in the updated manuscript:

Line 411-413: “*However, this is only true for gene variants acquired from closely related species and these fitness effects likely depend on the environmental conditions; e.g. the growth medium.*”

And this has also been highlighted in the concluding paragraph:

Line 421-423:

“*However, knowledge on antibiotic use, co-selection, regulatory or compensatory interactions in a range of hosts and growth conditions is needed for more accurate predictions⁵¹*”